# LATENT POINT COLLAPSE
# ON A LOW DIMENSIONAL EMBEDDING
# IN DEEP NEURAL NETWORK CLASSIFIERS

## ABSTRACT

The topological properties of latent representations play a critical role in determining the performance of deep neural network classifiers. In particular, the emergence of well-separated class embeddings in the latent space has been shown to improve both generalization and robustness. In this paper, we propose a method to induce the collapse of latent representations belonging to the same class into a single point, which enhances class separability in the latent space while making the network Lipschitz continuous. We demonstrate that this phenomenon, which we call *latent point collapse* (LPC), is achieved by adding a strong $L_2$ penalty on the penultimate-layer representations and is the result of a push-pull tension developed with the cross-entropy loss function. In addition, we show the practical utility of applying this compressing loss term to the latent representations of a low-dimensional linear penultimate-layer. LPC can be viewed as a stronger manifestation of *neural collapse* (NC): while NC entails that within-class representations converge around their class means, LPC causes these representations to collapse in absolute value to a single point. As a result, the network improvements typically associated with NC—namely better robustness and generalization—are even more pronounced when LPC develops.

## 1 INTRODUCTION

Deep neural networks (DNNs) excel in various tasks, but they often struggle with ensuring robust performance and reliable generalization. A key insight into addressing these challenges lies in understanding and controlling the geometry of the latent representations that DNNs learn. In particular, it has been observed that increasing the margin between classes, i.e., making classes more separable in the latent space, can yield significant gains in both robustness and generalization (1; 2; 3; 4). Indeed, the relationship between generalization and robustness is well-established in the literature (5; 6; 7; 8; 9).

DNNs naturally tend to improve the separation of different classes in the latent space during training, and this process occurs at a constant geometric rate (10). Such evolving separation manifests in the phenomenon of *neural collapse* (NC) (11; 12). While NC is prominently observed and analyzed in the penultimate-layer of DNN classifiers, its characteristics, propagation, and related phenomena have also been investigated in intermediate layers (13; 14; 15; 16; 17). This phenomenon typically occurs in overparameterized models during the terminal phase of training (TPT). Even after the point of zero training error, the network further refines the representations by increasing their relative distances in the latent space. In practice, this means that the class means in the penultimate-layer collapse to the vertices of an equiangular tight-frame simplex (ETFS). The occurrence of NC has been observed in large language models (18), in graph neural networks (19), with multivariate regression (20), and extensively investigated in unconstrained feature models (UFMs) (21; 22; 23; 24; 25; 26; 27; 28; 29; 30; 31; 32; 33; 34; 35) and in the mean-field regime (36). Nonetheless, perfect convergence to an ETFS is not always observed in practical scenarios (37).

NC implies convergence to a neural geometry that enhances separability in the latent space, ultimately improving generalization and robustness. In fact, it has been shown that these metrics continue to improve during the TPT (11), precisely when the latent representations approach an ETFS.

Subsequent research has revealed that NC also brings other benefits. For instance, NC has been linked to improved knowledge distillation (38), few-shot class incremental learning (39) and transfer learning (40; 41; 42; 43). Building on this connection, (44; 45; 46) use NC-based metrics to enhance the transferability of models. Another direction relates NC with out-of-distribution (OOD) detection (47; 48; 49; 50) and generalization (51).

A separate line of research for improving robustness in DNNs focuses on developing Lipschitz continuous networks, as Lipschitz constraints help ensure bounded responses to input perturbations (52; 53; 54; 55). Specifically, in Lipschitz networks, the smallest perturbation that can cause misclassification is inversely proportional to the Lipschitz constant (52; 56).

## 1.1 CONTRIBUTIONS

We introduce *latent point collapse* (LPC), a phenomenon in which penultimate-layer representations of each class converge to distinct points near the origin under **strong** $L_2$ regularization. Unlike NC, which permits unbounded representation growth and only achieves relative convergence, LPC enforces strict geometric confinement with provable Lipschitz continuity guarantees—a property that standard NC cannot ensure. The key distinction lies in the collapse metrics. While NC theory formally defines collapse as the within-class covariance $\Sigma_W \to 0$, practical implementations only achieve the weaker condition $\text{Tr}(\Sigma_W \Sigma_B^\dagger) \to 0$, where $\Sigma_B$ is the between-class covariance. This relative metric allows substantial within-class variance as long as between-class separation dominates. Our theoretical analysis proves that cross-entropy loss alone drives unbounded norm growth, preventing true collapse. In contrast, LPC achieves $\Sigma_W \to 0$ absolutely, with representations confined close to the origin. This confinement directly yields global Lipschitz continuity—a critical robustness property unattainable through standard NC. Our specific contributions are:

- **Discovery and theoretical characterization of latent point collapse:** We identify and rigorously analyze the emergent phenomenon of LPC, where strong $L_2$ regularization induces strict collapse of class representations to equilibrium points near the origin and ensures global Lipschitz continuity (Appendix A). We further demonstrate that LPC represents a stronger manifestation of standard NC properties, with faster convergence to ETFS structure (Appendix E).

- **Empirical validation and performance improvements:** We experimentally confirm LPC in practice, observing collapse points near the origin and improved Lipschitz properties (Section 3.1). We demonstrate remarkable improvements in robustness along with statistically significant gains in generalization (Section 3.2). Our approach uses a low-dimensional linear penultimate-layer that acts as a bottleneck, amplifying both LPC and performance.

- **Information bottleneck connection (Appendix B):** We establish that LPC naturally induces an information bottleneck in the penultimate-layer, providing an information-theoretic perspective on the observed generalization improvements.

- **Binary encoding (Appendix D):** We document an emergent phenomenon where penultimate latent representations converge to hypercube vertices.

## 1.2 RELATED WORKS

Our approach employs $L_2$ regularization on latent representations rather than network weights, a critical distinction from conventional regularization strategies. While weight-based penalties such as $L_0$ (57), $L_1$ (58), dropout (59), or weight decay are well-established, they do not induce the geometric collapse phenomenon we observe. By targeting penultimate-layer representations directly, we achieve feature compression without architectural modifications.

The application of $L_2$ penalties to latent representations is common in theoretical analyses of UFMs (27; 34; 24; 21; 25; 33; 32; 31; 30; 29; 28), where both weight and feature regularization were employed to establish existence of global optimizers. However, these works used minimal regularization coefficients (typically $\gamma \ll 1$) that served purely as mathematical convenience without affecting network behavior. Such weak regularization neither induces observable LPC nor provides practical benefits. In stark contrast, we employ extreme regularization strengths ($\gamma = 10^6$ in our experiments) that fundamentally alter the optimization landscape. Our theoretical analysis

(Appendix A) reveals that this creates a qualitatively different regime where the loss landscape becomes globally strongly convex. The strong $L_2$ penalty establishes a confining potential that counteracts the unbounded growth inherent to cross-entropy minimization. This tension between opposing forces—cross-entropy pushing representations outward while $L_2$ regularization pulls them toward the origin—induces LPC. It is not trivial that this severe confinement near the origin enhances rather than degrades performance, while simultaneously guaranteeing global Lipschitz continuity. The novelty of our work lies not in using regularization itself, but in the discovery and analysis of this dynamic.

Adding loss terms to intermediate layers has also appeared in the context of deep supervision (60; 61), where intermediate outputs are trained to match target labels. However, our approach differs by seeking to compress the volume of latent representations rather than providing additional supervision signals.

Various methods have been devised to enlarge inter-class margins, such as contrastive learning (62; 63; 64; 65; 66; 67; 68) and supervised contrastive learning (SupCon) (69), which pull together positive samples while pushing apart negative ones. Other techniques alter the loss function to reduce intra-class variance (70) or impose angular constraints (71; 72), e.g., CosFace (73) and ArcFace (74). These last two methods, ArcFace and CosFace, can be compared to our method in their simplicity, as they each introduce a single penalty term to the loss function to increase margins.

Our approach also makes the network Lipschitz continuous. Prior works on Lipschitz neural networks often rely on architectural constraints such as spectral norm regularization (75; 53; 54), orthogonal weight matrices (76; 77; 78), or norm-bound weights (79; 80), which can reduce model expressiveness or be computationally expensive. By contrast, our method imposes no specialized architectural constraints.

## 2 METHOD

Given a labeled dataset $\{\boldsymbol{x}_i, \bar{y}_i\}_{i=1}^N$, where $N$ denotes the number of training samples, we address the problem of multi-class classification using DNNs. We employ a DNN $\boldsymbol{f}(\boldsymbol{x})$ that learns a nonlinear mapping from input space to output space, approximating the underlying data distribution. DNNs consist of multiple layers arranged hierarchically, with each layer producing an intermediate latent representation. The network's output can be expressed as a composition of layer-wise transformations: $\boldsymbol{f}(\boldsymbol{x}) = \boldsymbol{f}^{(M)} \circ \boldsymbol{f}^{(M-1)} \circ \cdots \circ \boldsymbol{f}^{(1)}(\boldsymbol{x})$, where $M$ denotes the depth of the network.

For an input vector $\boldsymbol{x}$, the forward pass through the network can be conceptually divided into two stages. First, the nonlinear components transform the input into a high-dimensional latent representation $\boldsymbol{h}(\boldsymbol{x})$, corresponding to the output of the final hidden layer. Subsequently, a linear classifier maps this representation to the output space: $\boldsymbol{f}(\boldsymbol{x}) = \boldsymbol{W}\boldsymbol{h}(\boldsymbol{x}) + \boldsymbol{b}$, where $\boldsymbol{W}$ and $\boldsymbol{b}$ denote the weight matrix and bias vector of the classifier, respectively. The predicted class label $y$ is obtained by applying the softmax function to the network's output, yielding a probability distribution over classes that quantifies the likelihood of input $\boldsymbol{x}$ belonging to each class. The network parameters are optimized by minimizing the cross-entropy loss: $\mathcal{L}_{\text{CE}}(\boldsymbol{f}(\boldsymbol{x}), \bar{y}) = -\log \frac{\exp(f_{\bar{y}}(\boldsymbol{x}))}{\sum_j \exp(f_j(\boldsymbol{x}))}$, which measures the divergence between predicted probabilities and ground-truth labels.

We propose augmenting the architecture with an additional linear transformation preceding the classifier: $\boldsymbol{z} = \boldsymbol{W}_{\text{L2}}\boldsymbol{h}(\boldsymbol{x}) + \boldsymbol{b}_{\text{L2}}$. This layer functions as the penultimate representation, with final classification performed via: $\boldsymbol{f}(\boldsymbol{x}) = \boldsymbol{W}\boldsymbol{z} + \boldsymbol{b}$. Beyond the standard cross-entropy loss, we introduce an $L_2$ regularization term applied to the penultimate-layer: $\mathcal{L}_2(\boldsymbol{z}) = ||\boldsymbol{z}||^2$, where $|| \cdot ||$ denotes the Euclidean norm. The composite loss function becomes: $\mathcal{L} = \mathcal{L}_{\text{CE}} + \gamma \mathcal{L}_{\text{L2}}$, where $\gamma > 0$ is a regularization hyperparameter controlling the strength of the $L_2$ penalty. The phenomenon of LPC arises from the interplay between these competing objectives:

$$\mathcal{L} = -\log \frac{\exp((\boldsymbol{W}\boldsymbol{z} + \boldsymbol{b})_{\bar{y}})}{\sum_j \exp((\boldsymbol{W}\boldsymbol{z} + \boldsymbol{b})_j)} + \gamma ||\boldsymbol{z}||^2. \tag{1}$$

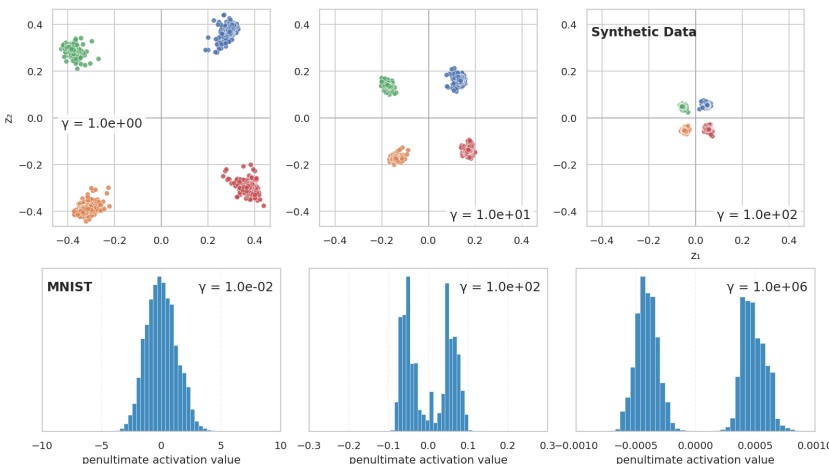

Figure 1: Latent point collapse progression on synthetic and MNIST datasets. *Top row:* 2D penultimate layer activations for a 4-class synthetic dataset at increasing $\gamma$ regularization strengths. Each color represents a different class. The progression illustrates the competing forces at play: the cross-entropy loss drives class separation, while the $\gamma\|\mathbf{z}\|^2$ penalty compresses representations toward the origin. As $\gamma$ increases, both the distance from the origin and the within-class spread decrease, yet class separability is preserved. *Bottom three rows:* Histograms of penultimate activations for MNIST classification across all penultimate nodes and datapoints. As $\gamma$ increases, the activation distribution becomes increasingly concentrated around zero. At high $\gamma$ values, the network responds to the extreme compression by converging toward binary encoding, with activations clustering at distinct points opposite to each other relative to the origin. Both experiments demonstrate how LPC regularization systematically reduces representation norms while maintaining discriminative power. Jupyter notebook to reproduce results is available in the linked repository.[4.4]

## 2.1 THEORETICAL CHARACTERIZATION AND MECHANISM

The *latent point collapse* phenomenon, illustrated in Figure 1, occurs when strong $L_2$ regularization applied to penultimate layer representations causes representations to collapse into distinct, class-specific points while maintaining separation between classes. LPC emerges from the interplay between competing optimization forces. We provide intuition here; rigorous proofs and complete theoretical characterization appear in Appendix A.

**The necessity of regularization.** The unconstrained feature model (UFM) framework (24; 34; 27) predicts that global minimizers of cross-entropy loss exhibit perfect NC—all within-class features converge to single points forming an ETFS. However, practical deep networks rarely reach such optima (37).

Our analysis reveals the fundamental obstacle: pure cross-entropy optimization exhibits an intrinsic instability. Its gradient contains an outward radial component that causes representations to grow unboundedly, preventing convergence to finite equilibria. We formalize this as a necessity result: without regularization ($\gamma = 0$), the outward radial component drives $\|\mathbf{z}\| \to \infty$, precluding any finite equilibria (Theorem A.8). As norms increase, softmax probabilities saturate and gradients vanish preventing the formation of any structured geometry.

Strong $L_2$ regularization resolves this instability by introducing an inward restoring force $-2\gamma\mathbf{z}$ that counteracts the outward push. At equilibrium, these competing forces balance according to:

$$\mathbf{z}^* = \frac{1}{2\gamma}\left(\mathbf{w}_{\bar{y}} - \sum_{i=1}^{K} p_i(\mathbf{z}^*)\mathbf{w}_i\right) \qquad (2)$$

This reveals that the equilibrium radius scales inversely with regularization strength: $\|\mathbf{z}^*\| \le M_W/\gamma$, where $M_W$ bounds classifier weight norms (Theorem A.4). Figure 2 (left) empirically validates this scaling.

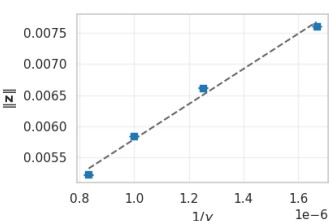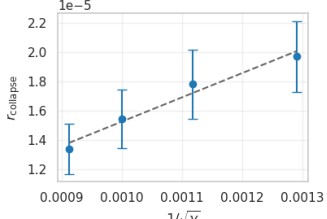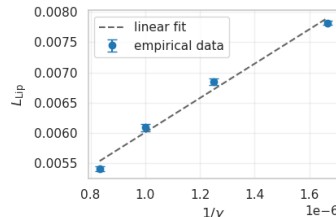

Figure 2: Scaling relationships between LPC regularization strength and penultimate layer geometry. CIFAR-10 trained as in Appendix C but with varying regularization strengths $\gamma_{max} \in \{6 \times 10^5, 8 \times 10^5, 10^6, 1.2 \times 10^6, 1.4 \times 10^6\}$ and fixed initial learning rate $\eta = 10^{-4}$. Blue points show empirical measurements with error bars indicating standard deviation across 10 samples. Dashed lines represent linear fits to the data. From left to right: Mean penultimate feature norm $\mathbb{E}[\|\mathbf{z}\|]$ as a function of $1/\gamma$. Mean within-class distance from class centroid $\mathbb{E}[\|\mathbf{z} - \boldsymbol{\mu}_{\bar{y}}\|]$ as a function of $1/\sqrt{\gamma}$. Full radius containing all penultimate representations $\max_i \|\mathbf{z}_i\|$ as a function of $1/\gamma$. Linear fits demonstrate strong scaling relationships with $R^2 = 0.9901$, $R^2 = 0.9688$, and $R^2 = 0.9857$, respectively. The tight linear relationships confirm theoretical predictions that LPC regularization systematically controls the geometric properties of learned representations.

**Convexification and collapse dynamics.** For sufficiently large regularization $\gamma > KM_W^2/2$, the Hessian becomes globally positive definite:

$$\nabla_{\mathbf{z}}^2 \mathcal{L}(\mathbf{z}) = \nabla_{\mathbf{z}}^2 \mathcal{L}_{\text{CE}}(\mathbf{z}) + 2\gamma\mathbf{I} \succeq (2\gamma - KM_W^2)\mathbf{I} \succ 0 \tag{3}$$

This *convexification* ensures global strong convexity, guaranteeing convergence to a unique global optimum.

Beyond establishing global convergence, strong regularization also controls local fluctuations. Near equilibrium, the dominant quadratic potential (curvature $\approx 2\gamma\mathbf{I}$) tightly constrains stochastic fluctuations from gradient noise. Under standard assumptions of bounded classifier weights and fixed gradient noise variance $\sigma^2$, our analysis establishes that steady-state variance decreases monotonically with $\gamma$. Specifically, the collapse radius—characterizing within-class spread—scales as $r_{\text{collapse}} = O(\sqrt{\sigma^2 d\eta/\gamma})$, where $d$ is representation dimensionality and $\eta$ is learning rate (Section A.3). Figure 2 (center) confirms this scaling.

Thus, $\gamma$ serves a dual role: it ensures convergence to the global optimum through landscape convexification while simultaneously controlling the tightness of intra-class clustering through increased curvature.

**Global Lipschitz continuity and robustness.** The bounded confinement directly yields global Lipschitz continuity. Since all representations lie within radius $O(M_W/\gamma)$ of the origin, the maximum possible distance between any two representations is bounded independently of input distance. We prove that for any inputs $\mathbf{x}_1, \mathbf{x}_2$:

$$\mathbb{E}[\|\mathbf{f}(\mathbf{x}_1) - \mathbf{f}(\mathbf{x}_2)\|] \leq \frac{2\sqrt{K}M_W^2}{\gamma} + O\left(\sqrt{\frac{\sigma^2 d\eta}{\gamma}}\right) \tag{4}$$

regardless of $\|\mathbf{x}_1 - \mathbf{x}_2\|$ (Theorem A.11). This uniform Lipschitz bound limits the sensitivity of network outputs to input perturbations, that is empirically demonstrated in Figure 2 (right).

Beyond confinement and collapse, the regularization induces a third geometric property in the terminal phase of training. When the network achieves high classification confidence ($p_{\bar{y}} \to 1$), the radial nature of the $L_2$ regularization decouples the angular dynamics from the radial dynamics, as the regularization term vanishes when projected onto the tangent space of the unit sphere. Strong regularization accelerates this alignment by reducing the equilibrium radius $r^* = O(M_W/\gamma)$, causing representations to progressively align with their corresponding classifier weight vectors: $\lim_{t\to\infty} \mathbf{z}(t)/\|\mathbf{z}(t)\| = \mathbf{w}_{\bar{y}}/\|\mathbf{w}_{\bar{y}}\|$ (Theorem A.7). This accelerated alignment is demonstrated by faster convergence of the NC property, as shown in Appendix E. These five key properties—necessity of regularization, bounded equilibria, tight collapse, weight alignment, and global Lipschitz continuity—are formally unified in Theorem A.13 (Appendix A).

## 2.2 BINARY ENCODING

The collapse points align with vertices of a hypercube inscribed within a hypersphere, as documented in Appendix C. At each penultimate-layer node, latent representations approximately assume one of two values, forming a binary encoding. One possible explanation is that extreme compression forces the network to maximize relative distances between collapse points *in each dimension* to maintain discriminability, naturally leading to symmetric arrangements around the origin characteristic of hypercube vertices.

## 3 EXPERIMENTS

We empirically demonstrate that our method promotes LPC in penultimate-layers and enhances classifier performance. Through ablation studies, we evaluate architectures differing in penultimate-layer dimensionality, linearity, $L_2$ regularization, and loss functions (see Appendix C for details).

**Architectures:** LPC (our method: linear penultimate-layer with $L_2$ regularization), LPC-NARROW/LPC-WIDE (low/high-dimensional variants), LPC-NOPEN ($L_2$ on backbone, no penultimate-layer), LINPEN/NONLINPEN (linear/non-linear penultimate-layer without $L_2$), NOPEN (baseline), NOPENWD (baseline with strong weight decay), SCL/ARCFACE (alternative losses), and LPC-SCL (hybrid).

Experiments on CIFAR-10, CIFAR-100 (81), and ImageNet-1K (82) using ResNet (83) and Wide ResNet (84).

## 3.1 LATENT POINT COLLAPSE: PROPERTIES AND LIPSCHITZ CONTINUITY

To investigate the occurrence of LPC in the penultimate-layer, we examine the within-class covariance $\boldsymbol{\Sigma}_W$, defined as

$$\boldsymbol{\Sigma}_W = \frac{1}{NP} \sum_{i=0}^{N-1} \sum_{p=0}^{P-1} \left( \boldsymbol{z}^{(i,p)} - \boldsymbol{\mu}^{(p)} \right) \left( \boldsymbol{z}^{(i,p)} - \boldsymbol{\mu}^{(p)} \right)^{\top}, \tag{5}$$

where $\boldsymbol{z}^{(i,p)}$ denotes the $i$-th latent representation with label $p$, and $\boldsymbol{\mu}^{(p)}$ represents the mean of all latent representations with label $p$. A vanishing $\boldsymbol{\Sigma}_W$ indicates that latent representations within each class collapse toward their respective class mean $\boldsymbol{\mu}^{(p)}$. We additionally analyze the mean norm $\frac{1}{P} \sum_{p=0}^{P-1} \|\boldsymbol{\mu}^{(p)}\|$, which quantifies the average distance of collapse points from the origin. Consistent with our theoretical analysis in Appendix A, we expect these collapse points to be located near the origin.

Table 1 presents the values of $\boldsymbol{\Sigma}_W$ and the mean norm at the final training epoch across various architectures. Notably, $\boldsymbol{\Sigma}_W$ approaches zero exclusively for architectures implementing an $L_2$ penalty on the penultimate-layer, confirming that same-class latent representations indeed collapse to single points in these models. The exceptionally small mean norm values across all LPC architectures corroborate our theoretical prediction that collapse points reside near the origin. Interestingly, we observe that strong weight decay alone, as employed in the NoPenWD architecture, does not induce collapse. The information bottleneck created by this point collapse phenomenon is further examined in Appendix B.

Beyond the geometric properties of the latent space, we analyze the resulting classification characteristics, particularly class separability and decision boundary stability. To quantify separability in the latent space, we introduce a *class separation ratio* $\mathcal{R}$: $\mathcal{R}^{(i,p)} = \frac{\min_{q \neq p} \|\boldsymbol{z}^{(i,p)} - \boldsymbol{\mu}^{(q)}\|}{\|\boldsymbol{z}^{(i,p)} - \boldsymbol{\mu}^{(p)}\|}$. This ratio measures the distance between a latent point and the nearest other-class centroid relative to its distance from its own-class centroid. Higher values indicate superior geometric separation between classes. As demonstrated in Table 1, architectures implementing the $L_2$ penalty on a penultimate linear layer achieve class separation ratios exceeding those of baseline methods by approximately two orders of magnitude, demonstrating markedly superior latent representation separation.

To evaluate classification boundary stability, we analyze the network's sensitivity to input perturbations through the function $g_y(\boldsymbol{x}) = f_y(\boldsymbol{x}) - \max_{j \neq y} f_j(\boldsymbol{x})$, where $f_y(\boldsymbol{x})$ denotes the logit of the true class $y$. This function captures the classification margin in the output space, with $g_y(\boldsymbol{x}) > 0$

Table 1: All values in the table represent the means and standard deviations obtained from different experiments. $\Sigma_W$: mean across all entries of the within-class covariance matrix; Norm Mean: mean of the $L_2$ norm of latent representations; $\mathcal{R}$: class separation ratio (distance margins, computed over a sample of 10000 entries in the training set); Avg. Grad Norm: Average $L_2$ Gradient Norm of the Logit Difference Function ($g_y(\boldsymbol{x})$) with respect to input $\boldsymbol{x}$.

| DATASET: CIFAR-10 | | | | |
|---|---|---|---|---|
| MODEL | $\Sigma_W$ | NORM MEAN | $\mathcal{R}$ | AVG. GRAD NORM |
| LPC | 2.55E-14 $\pm$ 1.38E-14 | 0.004 $\pm$ 0.001 | 195.94 $\pm$ 220.87 | 0.06 $\pm$ 0.03 |
| LPC-WIDE | 1.71E-14 $\pm$ 5.88E-15 | 0.003 $\pm$ 0.001 | 113.76 $\pm$ 126.59 | 0.15 $\pm$ 0.05 |
| LPC-NARROW | 5.70E-14 $\pm$ 1.05E-14 | 0.003 $\pm$ 0.001 | 241.28 $\pm$ 282.61 | 0.04 $\pm$ 0.03 |
| LPC-SCL | 9.89E-14 $\pm$ 4.50E-14 | 0.004 $\pm$ 0.001 | 98.36 $\pm$ 93.63 | 0.13 $\pm$ 0.04 |
| LPC-NOPEN | 1.33E-11 $\pm$ 6.21E-12 | 0.005 $\pm$ 0.001 | 31.77 $\pm$ 20.47 | 0.25 $\pm$ 0.26 |
| LINPEN | 1.10E-01 $\pm$ 4.00E-02 | 47.70 $\pm$ 7.89 | 2.90 $\pm$ 0.28 | 99.16 $\pm$ 10.61 |
| NONLINPEN | 4.08E-01 $\pm$ 1.42E-01 | 39.39 $\pm$ 7.17 | 2.87 $\pm$ 0.25 | 62.06 $\pm$ 9.66 |
| SCL | 8.02E-03 $\pm$ 4.13E-03 | 25.42 $\pm$ 2.37 | 7.77 $\pm$ 2.27 | 44.09 $\pm$ 1.81 |
| ARCFACE | 9.05E-02 $\pm$ 2.32E-02 | 22.81 $\pm$ 1.33 | 5.63 $\pm$ 0.26 | 78.91 $\pm$ 33.68 |
| NOPEN | 3.03E-02 $\pm$ 1.08E-02 | 13.88 $\pm$ 3.49 | 1.79 $\pm$ 0.10 | 84.55 $\pm$ 8.93 |
| NOPENWD | 2.32E-03 $\pm$ 2.50E-04 | 7.33 $\pm$ 0.19 | 2.30 $\pm$ 0.27 | 29.27 $\pm$ 1.69 |

| DATASET: CIFAR-100 | | | | |
|---|---|---|---|---|
| MODEL | $\Sigma_W$ | NORM MEAN | $\mathcal{R}$ | AVG. GRAD NORM |
| LPC | 2.62E-12 $\pm$ 9.22E-13 | 0.007 $\pm$ 0.001 | 100.84 $\pm$ 110.22 | 0.11 $\pm$ 0.02 |
| LPC-WIDE | 4.08E-12 $\pm$ 4.13E-12 | 0.008 $\pm$ 0.002 | 79.35 $\pm$ 81.92 | 0.22 $\pm$ 0.10 |
| LPC-NARROW | 5.84E-12 $\pm$ 3.37E-12 | 0.006 $\pm$ 0.000 | 116.34 $\pm$ 134.08 | 0.06 $\pm$ 0.00 |
| LPC-SCL | 3.94E-12 $\pm$ 2.83E-12 | 0.008 $\pm$ 0.001 | 54.93 $\pm$ 58.23 | 0.22 $\pm$ 0.01 |
| LPC-NOPEN | 3.39E-06 $\pm$ 5.21E-06 | 0.018 $\pm$ 0.018 | 7.44 $\pm$ 8.79 | 6.99 $\pm$ 10.90 |
| LINPEN | 5.75E-01 $\pm$ 1.34E-01 | 74.28 $\pm$ 8.41 | 1.46 $\pm$ 0.08 | 86.82 $\pm$ 4.42 |
| NONLINPEN | 3.50E+00 $\pm$ 2.89E-01 | 92.74 $\pm$ 1.87 | 1.52 $\pm$ 0.09 | 81.82 $\pm$ 2.32 |
| SCL | 2.87E-03 $\pm$ 1.35E-03 | 34.52 $\pm$ 5.34 | 5.23 $\pm$ 1.91 | 66.25 $\pm$ 9.16 |
| ARCFACE | 4.40E-02 $\pm$ 2.70E-03 | 43.27 $\pm$ 1.16 | 4.42 $\pm$ 0.31 | 109.57 $\pm$ 11.50 |
| NOPEN | 1.94E-02 $\pm$ 1.44E-02 | 25.08 $\pm$ 6.26 | 1.15 $\pm$ 0.02 | 70.81 $\pm$ 3.03 |
| NOPENWD | 8.48E-04 $\pm$ 1.71E-04 | 13.02 $\pm$ 0.69 | 1.26 $\pm$ 0.09 | 40.22 $\pm$ 1.7 |

| DATASET: IMAGENET | | | | |
|---|---|---|---|---|
| MODEL | $\Sigma_W$ | NORM MEAN | $\mathcal{R}$ | AVG. GRAD NORM |
| LPC | 4.00E-10 $\pm$ 7.32E-12 | 0.013 $\pm$ 0.000 | 13.92 $\pm$ 0.04 | 1.67 $\pm$ 0.01 |
| NOPEN | 1.32E-02 $\pm$ 4.33E-05 | 37.10 $\pm$ 0.09 | 1.12 $\pm$ 0.00 | 12.46 $\pm$ 0.18 |

indicating correct classification. We quantify network sensitivity by computing the average gradient norm $\max \|\nabla_{\boldsymbol{x}} g_y(\boldsymbol{x})\|$ over the entire dataset, which provides an empirical lower bound estimate of the Lipschitz constant for $g_y(\boldsymbol{x})$.

Table 1 reports this average gradient norm. Models employing the $L_2$ penalty on the penultimate-layer exhibit average gradient norms approximately two orders of magnitude lower than alternative configurations, indicating substantially reduced sensitivity to input perturbations and enhanced output margin stability. Notable exceptions include LPC-NOPEN configuration, which can exhibit elevated values, emphasizing how dimensionality reduction in the penultimate-layer contributes to network stability. This reduced sensitivity, coupled with enhanced class separation, demonstrates that our LPC method yield classifications that are both geometrically well-separated and exhibit stronger Lipschitz continuity, thereby providing improved theoretical guarantees on network behavior under perturbations.

It is important to note that our experiments on CIFAR-10 and CIFAR-100 were conducted largely in the TPT, where training accuracy has essentially converged. In contrast, ImageNet experiments did not reach full convergence of training set accuracy. Remarkably, the advantages of our method—including superior class separation and reduced gradient norms—are already manifest before full convergence,

Table 2: All values in the table represent the means and standard deviations obtained from different experiments. *DeepFool*: Norm of the minimal perturbation to cause a prediction change, divided by the norm of the input (85), averaged over 1000 test set samples. *PGD columns*: Adversarial accuracy under PGD attacks (86) with $\epsilon \in \{4/255, 8/255, 12/255\}$ for CIFAR and $\epsilon \in \{2/255, 4/255\}$ for ImageNet, evaluated on 1000 test set samples. PGD uses 100 iterations (50 for ImageNet) with 5 random restarts, step size $\alpha = \epsilon/4$, DLR loss, cosine schedule, and $\ell_\infty$ norm constraint. *Accuracy*: Classification accuracy on the testing set. For ImageNet, the generalization gap (Gen. Gap) is also included, representing the difference between training and testing accuracy.

| | | DATASET: CIFAR-10 | | | |
|---|---|---|---|---|---|
| MODEL | DEEPFOOL | PGD $\epsilon$=4/255 | PGD $\epsilon$=8/255 | PGD $\epsilon$=12/255 | ACCURACY |
| LPC | $1.227 \pm 0.442$ | $0.146 \pm 0.021$ | $0.034 \pm 0.011$ | $\mathbf{0.013} \pm 0.007$ | $94.86 \pm 0.08$ |
| LPC-WIDE | $0.788 \pm 0.439$ | $0.180 \pm 0.059$ | $0.033 \pm 0.025$ | $0.010 \pm 0.008$ | $94.73 \pm 0.04$ |
| LPC-NARROW | $\mathbf{1.597} \pm 0.442$ | $0.153 \pm 0.029$ | $\mathbf{0.040} \pm 0.018$ | $\mathbf{0.013} \pm 0.006$ | $94.90 \pm 0.13$ |
| LPC-SCL | $0.521 \pm 0.059$ | $0.170 \pm 0.025$ | $0.034 \pm 0.007$ | $0.009 \pm 0.004$ | $\mathbf{94.91} \pm 0.10$ |
| LPC-NOPEN | $0.693 \pm 0.278$ | $\mathbf{0.195} \pm 0.034$ | $0.014 \pm 0.006$ | $0.001 \pm 0.002$ | $94.86 \pm 0.09$ |
| LINPEN | $0.013 \pm 0.001$ | $0.000 \pm 0.000$ | $0.000 \pm 0.000$ | $0.000 \pm 0.000$ | $94.58 \pm 0.08$ |
| NONLINPEN | $0.015 \pm 0.001$ | $0.001 \pm 0.001$ | $0.000 \pm 0.000$ | $0.000 \pm 0.000$ | $94.50 \pm 0.05$ |
| SCL | $0.026 \pm 0.001$ | $0.012 \pm 0.009$ | $0.001 \pm 0.001$ | $0.000 \pm 0.000$ | $94.77 \pm 0.11$ |
| ARCFACE | $0.019 \pm 0.001$ | $0.031 \pm 0.015$ | $0.000 \pm 0.001$ | $0.000 \pm 0.000$ | $94.54 \pm 0.08$ |
| NOPEN | $0.013 \pm 0.001$ | $0.000 \pm 0.001$ | $0.000 \pm 0.000$ | $0.000 \pm 0.000$ | $94.57 \pm 0.14$ |
| NOPENWD | $0.016 \pm 0.001$ | $0.000 \pm 0.000$ | $0.000 \pm 0.000$ | $0.000 \pm 0.000$ | $94.25 \pm 0.04$ |

| | | DATASET: CIFAR-100 | | | |
|---|---|---|---|---|---|
| MODEL | DEEPFOOL | PGD $\epsilon$=4/255 | PGD $\epsilon$=8/255 | PGD $\epsilon$=12/255 | ACCURACY |
| LPC | $0.399 \pm 0.022$ | $0.065 \pm 0.019$ | $0.013 \pm 0.007$ | $0.004 \pm 0.002$ | $77.64 \pm 0.17$ |
| LPC-WIDE | $0.369 \pm 0.015$ | $0.076 \pm 0.019$ | $0.015 \pm 0.004$ | $0.002 \pm 0.002$ | $77.75 \pm 0.24$ |
| LPC-NARROW | $\mathbf{0.470} \pm 0.026$ | $0.040 \pm 0.005$ | $0.004 \pm 0.002$ | $0.000 \pm 0.000$ | $77.29 \pm 0.18$ |
| LPC-SCL | $0.192 \pm 0.024$ | $0.049 \pm 0.005$ | $0.007 \pm 0.001$ | $0.003 \pm 0.001$ | $\mathbf{78.17} \pm 0.23$ |
| LPC-NOPEN | $0.151 \pm 0.036$ | $\mathbf{0.135} \pm 0.049$ | $\mathbf{0.034} \pm 0.018$ | $\mathbf{0.008} \pm 0.005$ | $77.27 \pm 0.36$ |
| LINPEN | $0.007 \pm 0.000$ | $0.001 \pm 0.001$ | $0.000 \pm 0.000$ | $0.000 \pm 0.000$ | $76.69 \pm 0.17$ |
| NONLINPEN | $0.007 \pm 0.000$ | $0.000 \pm 0.001$ | $0.000 \pm 0.000$ | $0.000 \pm 0.000$ | $76.38 \pm 0.30$ |
| SCL | $0.011 \pm 0.000$ | $0.004 \pm 0.002$ | $0.000 \pm 0.000$ | $0.000 \pm 0.000$ | $78.00 \pm 0.15$ |
| ARCFACE | $0.012 \pm 0.000$ | $0.105 \pm 0.023$ | $0.013 \pm 0.004$ | $0.001 \pm 0.002$ | $77.21 \pm 0.14$ |
| NOPEN | $0.007 \pm 0.000$ | $0.001 \pm 0.001$ | $0.000 \pm 0.000$ | $0.000 \pm 0.000$ | $76.83 \pm 0.15$ |
| NOPENWD | $0.007 \pm 0.000$ | $0.001 \pm 0.001$ | $0.000 \pm 0.000$ | $0.000 \pm 0.000$ | $76.97 \pm 0.14$ |

| | | DATASET: IMAGENET | | | |
|---|---|---|---|---|---|
| MODEL | DEEPFOOL | PGD $\epsilon$=2/255 | PGD $\epsilon$=4/255 | ACCURACY | GEN. GAP |
| LPC | $\mathbf{0.019} \pm 0.001$ | $\mathbf{0.045} \pm 0.005$ | $\mathbf{0.002} \pm 0.001$ | $\mathbf{74.48} \pm 0.05$ | $\mathbf{16.74} \pm 0.14$ |
| NOPEN | $0.002 \pm 0.000$ | $0.000 \pm 0.000$ | $0.000 \pm 0.000$ | $72.33 \pm 0.12$ | $24.08 \pm 0.12$ |

suggesting that these benefits emerge early in the optimization process rather than solely as a consequence of extended training.

Our experiments further reveal that the collapse points align with vertices of a hypercube, as detailed in Appendix D. Additionally, in Appendix E, we demonstrate that LPC enhances metrics associated with NC. Given that our training on CIFAR-10 and CIFAR-100 predominantly occurred in the TPT regime, these improvements complement and extend beyond those typically associated with standard NC.

### 3.2 ROBUSTNESS AND GENERALIZATION

Table 2 presents the magnitude of minimal perturbations required to induce misclassification, quantified using the DeepFool algorithm (85). Our results reveal a striking enhancement in network

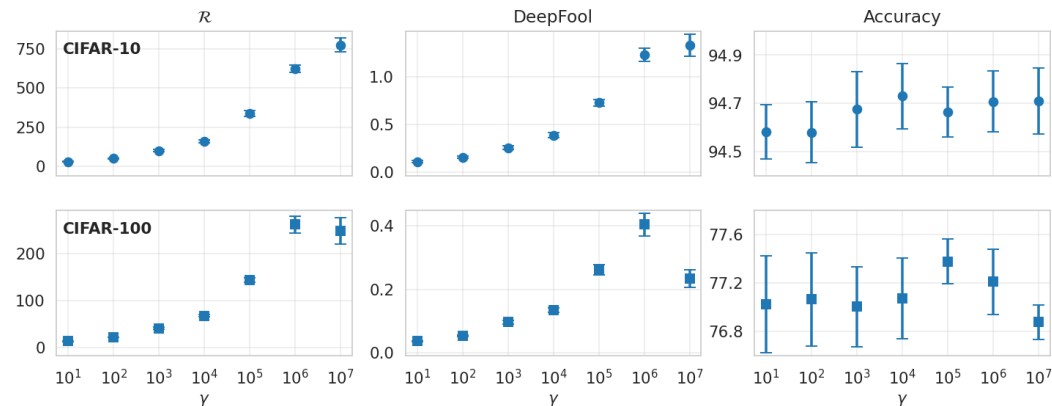

Figure 3: Effect of LPC regularization strength on classification performance and robustness. The figure shows accuracy, DeepFool robustness, and distance ratio $\mathcal{R}$ as functions of the LPC regularization parameter $\gamma$ for CIFAR-10 (top row) and CIFAR-100 (bottom row) datasets. Results are averaged across multiple independent runs with error bars indicating standard deviation. As $\gamma$ increases from $10^1$ to $10^6$, both geometric margins and robustness improve significantly while classification accuracy remains stable. CIFAR-10 maintains stable accuracy across the entire range ($\gamma \leq 10^7$), while CIFAR-100 shows peak accuracy at $\gamma = 10^5$ with degradation emerging only at the extreme value $\gamma = 10^7$.

robustness when $L_2$ regularization is applied to the penultimate-layer—achieving improvements exceeding two orders of magnitude, particularly pronounced when the penultimate-layer is linear. This robustness enhancement extends to adversarial attacks: LPC models maintain non-trivial adversarial accuracy under PGD attacks (86) (e.g., 14.6% at $\epsilon = 4/255$ on CIFAR-10, 6.5% at $\epsilon = 4/255$ on CIFAR-100, and 4.5% at $\epsilon = 2/255$ on ImageNet), while baseline models exhibit essentially zero robustness under comparable perturbation budgets. While alternative regularization techniques, including SCL and ArcFace, also demonstrate improved robustness relative to the baseline, their gains remain substantially more modest, with most achieving near-zero PGD robustness.

Beyond robustness improvements, LPC exhibits significant regularization effects that enhance generalization performance. Notably, LPC achieves generalization performance comparable to or exceeding state-of-the-art regularization methods such as SCL. The combination of LPC with SCL yields the highest generalization performance on both CIFAR-10 and CIFAR-100, demonstrating the potential for synergistic improvements when integrating LPC with existing techniques. The generalization improvements are particularly remarkable on ImageNet, where LPC reduces the generalization gap from 24.08% to 16.74%—a relative reduction of over 30%. These improvements are robust to the choice of regularization strength (Figure 3).

Our architectural analysis further reveals practical insights regarding the penultimate-layer design. Specifically, we observe that lower-dimensional penultimate-layers yield substantially superior robustness as measured by DeepFool, with the LPC-Narrow configuration achieving the highest scores across both CIFAR datasets. Interestingly, adversarial robustness under PGD attacks exhibits a different pattern: the LPC-NoPen variant demonstrates the highest adversarial accuracy. This suggests that the optimal architectural configuration may depend on the specific threat model under consideration.

## 4 DISCUSSION AND LIMITATIONS

### 4.1 DISCUSSION

The LPC phenomenon dramatically enhances the separability of latent representations, yielding remarkable improvements in robustness. These enhancements stem directly from superior class separability and a fundamentally stronger form of collapse than traditional NC. While NC ensures that within-class representations converge around their class means (achieving small within-class

variance relative to between-class variance), LPC enforces absolute collapse to single points. We demonstrate particular effectiveness when applying this regularization to low-dimensional linear penultimate-layers, where the dimensional bottleneck amplifies both the collapse effect and its benefits. A key distinction is that our approach inherently guarantees global Lipschitz continuity without architectural constraints—a property unattainable through standard NC.

Our ImageNet experiments reveal a notable aspect of LPC: its benefits manifest even before reaching the TPT. Unlike CIFAR experiments conducted primarily in the TPT regime, ImageNet training did not achieve full convergence, yet LPC reduced the generalization gap substantially—a relative improvement of over one-third. The early emergence of enhanced class separation and reduced gradient norms indicates that the geometric benefits of LPC develop progressively with the optimization dynamics, providing practical advantages even in computationally constrained scenarios where full convergence is infeasible.

### 4.2 LIMITATIONS AND OUTLOOK

Our study focuses exclusively on balanced datasets, leaving unexplored the interaction between LPC and class imbalance. Recent investigations of NC under imbalanced conditions (30; 31; 32; 87) suggest potentially complex dynamics that warrant future investigation. Whether LPC's absolute collapse provides advantages or poses challenges in imbalanced scenarios remains an open question.

While we demonstrate LPC under cross-entropy loss with $L_2$ regularization, NC has been observed with various loss functions (12; 88). Exploring alternative loss formulations that might induce or enhance LPC could reveal more efficient training procedures or stronger guarantees.

Our analysis primarily examines geometric and robustness metrics established in the original NC literature (11). However, NC enhances numerous other properties including knowledge distillation (38), few-shot learning (39), transfer learning (40; 41), and out-of-distribution detection (47; 48). The stronger absolute collapse of LPC may amplify these benefits, presenting promising avenues for investigation.

The empirically observed binary encoding phenomenon (Appendix D), wherein collapse points align precisely with hypercube vertices inscribed within a hypersphere, is an intriguing finding. Under extreme $L_2$ regularization, the network spontaneously organizes class representations into a discrete binary scheme where each penultimate node assumes one of two values, effectively creating a $\{-1, +1\}^d$ encoding. Understanding this spontaneous quantization could bridge connections to binary neural networks and discrete representation learning, though the precise mechanisms driving this phenomenon remain an open theoretical question warranting further investigation.

### 4.3 CONCLUSION

We introduce *latent point collapse*, a phenomenon where strong $L_2$ regularization applied to penultimate-layer representations causes the features within each class to collapse to a single point near the origin. This simple modification—achieved by adding a regularized, low-dimensional linear penultimate-layer—dramatically improves model robustness while also yielding statistically significant gains in generalization.

The method's simplicity and effectiveness make it immediately practical. By inducing a stronger form of *neural collapse* with provable Lipschitz guarantees, LPC provides principled bounds on adversarial robustness while maintaining or improving classification accuracy. The early emergence of benefits, demonstrated in our ImageNet experiments, indicates that LPC's advantages are accessible even without complete training convergence, enhancing its practical applicability in resource-constrained settings.

### 4.4 REPRODUCIBILITY STATEMENT

Code to reproduce our results is available online in the linked repository.[1] All experimental details are provided in Appendix C.

---

[1] https://anonymous.4open.science/r/lpc-0CEB

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

## A  THEORETICAL ANALYSIS OF LATENT POINT COLLAPSE

This appendix presents a rigorous theoretical analysis of how strong $L_2$ regularization induces LPC in DNN classifiers, establishing both its necessity for achieving theoretical optima and its sufficiency for ensuring global Lipschitz continuity. We employ the UFM framework to demonstrate that a large regularization parameter $\gamma$ fundamentally reshapes the optimization landscape and dynamics of penultimate-layer representations.

Our analysis proceeds through a systematic characterization of the interplay between cross-entropy loss and $L_2$ regularization. We first establish equilibrium conditions (Section A.2), proving that regularization confines representations to a ball of radius $O(M_W/\gamma)$, where $M_W$ bounds the classifier weights. We then analyze the local Hessian structure (Section A.3) to show that for $\gamma > K M_W^2/2$, the loss landscape becomes globally strongly convex. This convexification enables us to derive tight bounds on the steady-state distribution of representations, establishing that same-class features concentrate within radius $r_{\text{collapse}} = O(\sqrt{\sigma^2 d\eta/\gamma})$ of their class-specific equilibria, whose size reflects a balance between the noise level $\sigma$, the dimensionality of the latent space $d$, the magnitude of learning rate $\eta$, and the strength of the confining potential $\gamma$.

The geometric implications of this confinement are explored through our alignment analysis (Section A.4), where we prove that in the TPT—characterized by high classification confidence—representations not only collapse but also align with their corresponding classifier weight vectors. This alignment emerges naturally from the decoupling of radial and angular dynamics under strong regularization, without requiring explicit geometric constraints. Crucially, we establish the necessity of this mechanism by demonstrating (Section A.5) that without regularization, gradient-based optimization of cross-entropy loss exhibits a pathological behavior: the steepest descent direction contains an outward radial component, leading to unbounded norm growth and gradient saturation that prevents convergence to any finite equilibrium.

These results culminate in our main theorem establishing that strong $L_2$ regularization applied specifically to the penultimate-layer induces a stronger manifestation of NC characterized by simultaneous confinement around the origin and class-specific point collapse. Most significantly, we prove that this mechanism yields global Lipschitz continuity that is independent of input distance, a property that emerges naturally from the bounded collapse around the origin.

### A.1  SETUP AND PRELIMINARIES

We analyze penultimate-layer representations using the unconstrained feature model (UFM) framework (24; 34; 27). In this framework, features after the penultimate-layer are treated as free optimization variables $\boldsymbol{z} \in \mathbb{R}^d$ (disconnected from the input samples), allowing us to study their optimization dynamics independently. The final linear classifier has weights $\boldsymbol{W} \in \mathbb{R}^{K \times d}$ and biases $\boldsymbol{b} \in \mathbb{R}^K$, yielding logits $\boldsymbol{l} = \boldsymbol{W}\boldsymbol{z} + \boldsymbol{b}$.

Previous work (24; 34) established that for balanced datasets with bounded representations and classifier weights, global minimizers of cross-entropy loss exhibit NC. This phenomenon involves three key properties: (1) all features within a class converge to a single point, (2) the $K$ class means form an ETFS, and (3) these means align with their corresponding classifier weights. However, reaching such global optima in deep networks with millions of parameters is practically infeasible (37).

Our analysis investigates how strong $L_2$ regularization with coefficient $\gamma$ drives the system toward this theoretical ideal. We analyze the combined loss:

$$\mathcal{L}(\boldsymbol{z}) = \mathcal{L}_{\text{CE}}(\boldsymbol{z}) + \gamma\|\boldsymbol{z}\|^2 \tag{6}$$

where $\mathcal{L}_{\text{CE}}(\boldsymbol{z}) = -\log p_{\bar{y}}(\boldsymbol{z})$ is the cross-entropy loss with softmax probabilities $p_i(\boldsymbol{z}) = \frac{\exp(\ell_i)}{\sum_{j=1}^{K} \exp(\ell_j)}$, and $\gamma > 0$ is the regularization strength.

**Assumption A.1** (Bounded Classifier Weights). *The classifier weight vectors satisfy $\|\boldsymbol{w}_i\| \leq M_W$ for all classes $i \in [K]$, where $M_W < \infty$. This mild condition holds under standard training practices with weight decay, which keeps classifier weights bounded.*

We consider a general class of stochastic optimizers that update representations according to:

$$\boldsymbol{z}_{t+1} = \boldsymbol{z}_t - \eta_t \boldsymbol{P}_t(\nabla_{\boldsymbol{z}}\mathcal{L}(\boldsymbol{z}_t) + \mathbf{a}_t) \tag{7}$$

where $\eta_t$ is the learning rate, $\boldsymbol{P}_t$ is a preconditioning matrix (identity for SGD, diagonal adaptive for Adam/AdamW), and $\mathbf{a}_t$ is gradient noise from mini-batch sampling with $\mathbb{E}[\mathbf{a}_t|\boldsymbol{z}_t] = \mathbf{0}$.

**Assumption A.2** (Optimizer Properties). *The optimizer satisfies:*

1. ***Bounded preconditioning:*** $\lambda_{\min}\boldsymbol{I} \preceq \boldsymbol{P}_t \preceq \lambda_{\max}\boldsymbol{I}$ *for some* $0 < \lambda_{\min} \leq \lambda_{\max} < \infty$

2. ***Bounded noise:*** *The noise covariance matrix is bounded:* $\mathbb{E}[\mathbf{a}_t\mathbf{a}_t^T|\boldsymbol{z}_t] \preceq \sigma^2\boldsymbol{I}$ *for some constant* $\sigma^2 > 0$. *This implies* $\mathbb{E}[\|\mathbf{a}_t\|^2|\boldsymbol{z}_t] \leq d\sigma^2$.

3. ***Convergence:*** *The learning rate schedule ensures convergence to a neighborhood of local minima*

These assumptions are mild and satisfied by standard optimizers (SGD, Adam, AdamW) with appropriate hyperparameters. We will leverage these properties throughout our analysis to establish bounds on the behavior of representations under stochastic optimization.

## A.2 EQUILIBRIUM ANALYSIS

We first characterize the equilibrium points that arise from the interplay between cross-entropy loss and $L_2$ regularization.

**Definition A.3** (Stochastic Equilibrium). *A point* $\boldsymbol{z}^*$ *is a stochastic equilibrium if* $\mathbb{E}[\boldsymbol{z}_{t+1}|\boldsymbol{z}_t = \boldsymbol{z}^*] = \boldsymbol{z}^*$ *under the update rule equation 7.*

For optimizers with positive definite preconditioning matrices (Assumption A.2), this condition requires $\nabla_{\boldsymbol{z}}\mathcal{L}(\boldsymbol{z}^*) = \mathbf{0}$.

At a stochastic equilibrium $\boldsymbol{z}^*$, for a sample with true label $\bar{y}$, the gradient vanishes:

$$\nabla_{\boldsymbol{z}}\mathcal{L}(\boldsymbol{z}^*) = \sum_{i=1}^{K} p_i(\boldsymbol{z}^*)\boldsymbol{w}_i - \boldsymbol{w}_{\bar{y}} + 2\gamma\boldsymbol{z}^* = \mathbf{0} \tag{8}$$

Rearranging yields an explicit characterization:

$$\boldsymbol{z}^* = \frac{1}{2\gamma}\left(\boldsymbol{w}_{\bar{y}} - \sum_{i=1}^{K} p_i(\boldsymbol{z}^*)\boldsymbol{w}_i\right) \tag{9}$$

This equation reveals that the equilibrium position is scaled by a control parameter $\gamma$.

**Theorem A.4** (Bounded Equilibrium). *Any stochastic equilibrium* $\boldsymbol{z}^*$ *satisfies:*

$$\|\boldsymbol{z}^*\| \leq \frac{M_W}{\gamma} \tag{10}$$

*Proof.* From Equation 9, taking norms:

$$2\gamma\|\boldsymbol{z}^*\| = \left\|\boldsymbol{w}_{\bar{y}} - \sum_{i=1}^{K} p_i(\boldsymbol{z}^*)\boldsymbol{w}_i\right\| \tag{11}$$

$$= \left\|(1 - p_{\bar{y}}(\boldsymbol{z}^*))\boldsymbol{w}_{\bar{y}} - \sum_{i\neq\bar{y}} p_i(\boldsymbol{z}^*)\boldsymbol{w}_i\right\| \tag{12}$$

$$\leq (1 - p_{\bar{y}}(\boldsymbol{z}^*))\|\boldsymbol{w}_{\bar{y}}\| + \sum_{i\neq\bar{y}} p_i(\boldsymbol{z}^*)\|\boldsymbol{w}_i\| \tag{13}$$

$$\leq M_W\left[(1 - p_{\bar{y}}(\boldsymbol{z}^*)) + \sum_{i\neq\bar{y}} p_i(\boldsymbol{z}^*)\right] \tag{14}$$

$$= 2M_W(1 - p_{\bar{y}}(\boldsymbol{z}^*)) \tag{15}$$

The general bound follows since $1 - p_{\bar{y}}(\boldsymbol{z}^*) \leq 1$. $\qquad\square$

This theorem establishes that $L_2$ regularization creates a confining potential that prevents representation explosion, with the confinement radius inversely proportional to $\gamma$.

### A.3 HESSIAN ANALYSIS AND STOCHASTIC DYNAMICS

To understand how representations behave near equilibrium under stochastic optimization, we analyze the local curvature of the loss landscape. The Hessian of the combined loss is:

$$\nabla_z^2 \mathcal{L}(z) = \nabla_z^2 \mathcal{L}_{\text{CE}}(z) + \nabla_z^2(\gamma\|z\|^2) \tag{16}$$

The Hessian of the $L_2$ regularization term is:

$$\nabla_z^2(\gamma\|z\|^2) = 2\gamma I \tag{17}$$

The cross-entropy Hessian takes the form:

$$\nabla_z^2 \mathcal{L}_{\text{CE}}(z) = W^T(\text{diag}(p) - pp^T)W \tag{18}$$

with $p = [p_1(z), \ldots, p_K(z)]^T$ being the softmax probability vector.

**Lemma A.5** (Bounded Cross-Entropy Hessian). *Under Assumption A.1, the spectral norm of the cross-entropy Hessian satisfies:*

$$\|\nabla_z^2 \mathcal{L}_{CE}(z)\| \le K M_W^2 \tag{19}$$

*Proof.* The matrix $D = \text{diag}(p) - pp^T$ is the covariance matrix of a categorical distribution with spectral norm $\|D\|_2 \le 1$. The spectral norm of the Hessian is:

$$\|\nabla_z^2 \mathcal{L}_{\text{CE}}(z)\|_2 = \|W^T D W\|_2 \le \|W^T\|_2 \|D\|_2 \|W\|_2 = \|W\|_2^2 \tag{20}$$

The spectral norm of $W$ is bounded by its Frobenius norm:

$$\|W\|_2^2 \le \|W\|_F^2 = \sum_{i=1}^K \|w_i\|_2^2 \le K M_W^2 \tag{21}$$

where the last inequality follows from Assumption A.1. $\qquad\square$

This lemma shows that the cross-entropy contribution to the Hessian is bounded, allowing the $L_2$ regularization term to dominate for large $\gamma$.

#### A.3.1 EQUILIBRIUM NEIGHBORHOOD DYNAMICS

The bounded Hessian structure established in Lemma A.5 enables us to analyze how representations behave near equilibrium points under stochastic optimization. The regularization parameter $\gamma$ fundamentally reshapes the loss landscape by eliminating local minima that could trap the optimization process.

Consider the Hessian at an arbitrary point $z$:

$$\nabla_z^2 \mathcal{L}(z) = \nabla_z^2 \mathcal{L}_{\text{CE}}(z) + 2\gamma I \tag{22}$$

When $\gamma$ is sufficiently large, specifically when $\gamma > \frac{K M_W^2}{2}$, guarantees that the Hessian is globally positive definite. This implies that the loss landscape becomes strongly convex. This elimination of local minima ensures that stochastic optimization converges to the desired equilibrium rather than getting trapped in suboptimal configurations.

Near an equilibrium point $z^*$, the dynamics are governed by the local curvature. For large $\gamma$ satisfying $\gamma \gg K M_W^2$, the Hessian becomes dominated by the regularization term:

$$H = \nabla_z^2 \mathcal{L}(z^*) \approx 2\gamma I \tag{23}$$

This creates a strong, isotropic quadratic potential around the equilibrium, analogous to a harmonic oscillator with spring constant $k = 2\gamma$. Under stochastic optimization, representations do not converge to exact points but rather to steady-state distributions around equilibria. The concentration of these distributions is determined by the balance between the confining potential controlled by $\gamma$ and the stochastic excitation from gradient noise.

Consider the linearized dynamics around equilibrium:

$$\boldsymbol{d}_{t+1} = (\boldsymbol{I} - \eta \boldsymbol{P}_t \boldsymbol{H})\boldsymbol{d}_t - \eta \boldsymbol{P}_t \mathbf{a}_t \tag{24}$$

where $\boldsymbol{d}_t = \boldsymbol{z}_t - \boldsymbol{z}^*$ is the deviation from equilibrium, $\boldsymbol{P}_t$ is the optimizer's preconditioning matrix, and $\mathbf{a}_t$ represents gradient noise.

For the simplified case where $\boldsymbol{H} = 2\gamma \boldsymbol{I}$ and uniform preconditioning $\boldsymbol{P}_t = \bar{\lambda} \boldsymbol{I}$, the steady-state variance of each component satisfies:

$$\mathbb{E}[\delta_{i,\infty}^2] = \frac{\eta \bar{\lambda} \sigma^2}{4\gamma(1 - \eta\gamma\bar{\lambda})} \tag{25}$$

This expression reveals that for any fixed learning rate $\eta$ satisfying the stability condition $\eta < \frac{1}{\gamma\bar{\lambda}}$, the steady-state variance decreases monotonically as $\gamma$ increases. Taking the derivative with respect to $\gamma$ yields:

$$\frac{\partial}{\partial\gamma}\mathbb{E}[\delta_{i,\infty}^2] = -\frac{\eta\bar{\lambda}\sigma^2}{4\gamma^2(1 - \eta\gamma\bar{\lambda})^2} < 0 \tag{26}$$

confirming that increasing $\gamma$ always reduces the variance of representations around their equilibria.

The total expected squared deviation across all $d$ dimensions is:

$$\mathbb{E}[\|\boldsymbol{d}_\infty\|^2] = \sum_{i=1}^{d} \mathbb{E}[\delta_{i,\infty}^2] = \frac{d\eta\bar{\lambda}\sigma^2}{4\gamma(1 - \eta\gamma\bar{\lambda})} \tag{27}$$

For small learning rates satisfying $\eta\gamma\bar{\lambda} \ll 1$ (which is typically required for stability), we can approximate:

$$\mathbb{E}[\|\boldsymbol{d}_\infty\|^2] \approx \frac{d\eta\bar{\lambda}\sigma^2}{4\gamma} \tag{28}$$

Therefore, the root mean square deviation, which characterizes the typical collapse radius, is:

$$r_{\text{collapse}} = \sqrt{\mathbb{E}[\|\boldsymbol{d}_\infty\|^2]} \approx \frac{\sigma}{2}\sqrt{\frac{d\eta\bar{\lambda}}{\gamma}} = O\left(\sqrt{\frac{\sigma^2 d\eta}{\gamma}}\right) \tag{29}$$

where the last equality follows since $\bar{\lambda}$ and $\sigma$ are $O(1)$ constants independent of $\gamma$ and $\eta$.

This analysis reveals that a large regularization parameter $\gamma$ serves a crucial dual purpose. First, it convexifies the optimization landscape. For $\gamma > \frac{KM_W^2}{2}$, the regularization guarantees global strong convexity. Second, it induces feature collapse. The strong quadratic potential not only creates this single basin of attraction but also tightly confines representations within it, counteracting the stochasticity from gradient noise. This results in a steady-state variance that decreases monotonically with $\gamma$, leading to a tighter clustering of same-class representations.

Consequently, under a stable learning rate condition ($\eta < \frac{1}{\gamma\bar{\lambda}}$), the optimizer is guaranteed to converge to a neighborhood of the global optimum whose radius, $r_{\text{collapse}} = O(\sqrt{\sigma^2 d\eta/\gamma})$, is explicitly controlled by the regularization strength. This provides a principled explanation for the efficacy of strong $L_2$ regularization in achieving NC, as it ensures both convergence to the correct geometric configuration and control over the degree of intra-class feature concentration.

### A.4 ALIGNMENT CONVERGENCE ANALYSIS IN THE TERMINAL PHASE

In the TPT, when samples are well-classified with high confidence, we analyze how strong regularization induces alignment between representations and their corresponding classifier weights. Throughout this subsection, we assume the network has reached the TPT where $p_{\bar{y}}(\boldsymbol{z}) \approx 1$ for all samples in the training set.

To analyze this process, we consider the gradient flow dynamics $\dot{\boldsymbol{z}} = -\nabla_{\boldsymbol{z}}\mathcal{L}(\boldsymbol{z})$, a valid approximation for small learning rates. We decompose the dynamics into radial and angular components. Let $r(t) = \|\boldsymbol{z}(t)\|$ and $\boldsymbol{u}(t) = \boldsymbol{z}(t)/r(t)$ denote the representation's magnitude and direction, respectively.

**Definition A.6** (Representation-Weight Alignment). *For a representation $\boldsymbol{z}$ with direction $\boldsymbol{u} = \boldsymbol{z}/\|\boldsymbol{z}\|$ and true label $\bar{y}$, the alignment with the corresponding classifier weight is:*

$$a(t) = \boldsymbol{u}(t)^T \hat{\boldsymbol{w}}_{\bar{y}} = \frac{\boldsymbol{z}(t)^T \boldsymbol{w}_{\bar{y}}}{\|\boldsymbol{z}(t)\| \|\boldsymbol{w}_{\bar{y}}\|} \tag{30}$$

*where $\hat{\boldsymbol{w}}_{\bar{y}} = \boldsymbol{w}_{\bar{y}}/\|\boldsymbol{w}_{\bar{y}}\|$ is the normalized classifier weight. Note that $a(t) \in [-1, 1]$ by the Cauchy-Schwarz inequality, with $a(t) = 1$ indicating perfect alignment.*

We previously discussed that the $L_2$ term $2\gamma\boldsymbol{z}$ creates a strong restoring force in the radial direction, causing $r(t)$ to converge to an equilibrium radius $r^* = O(M_W/\gamma)$.

**Theorem A.7** (Alignment under $L_2$ Regularization). *In the terminal phase of training where $p_{\bar{y}}(\boldsymbol{z}) \approx 1$, under gradient flow dynamics with sufficiently large $\gamma$, the representation direction converges to the classifier weight direction:*

$$\lim_{t \to \infty} \frac{\boldsymbol{z}(t)}{\|\boldsymbol{z}(t)\|} = \frac{\boldsymbol{w}_{\bar{y}}}{\|\boldsymbol{w}_{\bar{y}}\|} \tag{31}$$

*Proof.* The angular velocity of the unit vector $\boldsymbol{u}$ is given by projecting the velocity $\dot{\boldsymbol{z}}$ onto the tangent space of the unit sphere:

$$\dot{\boldsymbol{u}} = \frac{1}{r}(\boldsymbol{I} - \boldsymbol{u}\boldsymbol{u}^T)\dot{\boldsymbol{z}} = -\frac{1}{r}(\boldsymbol{I} - \boldsymbol{u}\boldsymbol{u}^T)\nabla_{\boldsymbol{z}}\mathcal{L}(\boldsymbol{z}) \tag{32}$$

The gradient is $\nabla_{\boldsymbol{z}}\mathcal{L}(\boldsymbol{z}) = (\sum_i p_i\boldsymbol{w}_i - \boldsymbol{w}_{\bar{y}}) + 2\gamma\boldsymbol{z}$. Since $(\boldsymbol{I} - \boldsymbol{u}\boldsymbol{u}^T)\boldsymbol{z} = \boldsymbol{0}$, the regularization term does not affect the angular dynamics:

$$\dot{\boldsymbol{u}} = \frac{1}{r}(\boldsymbol{I} - \boldsymbol{u}\boldsymbol{u}^T)(\boldsymbol{w}_{\bar{y}} - \sum_i p_i\boldsymbol{w}_i) \tag{33}$$

The rate of change of alignment $a(t) = \boldsymbol{u}^T\hat{\boldsymbol{w}}_{\bar{y}}$ can be derived explicitly. We have:

$$\dot{a} = \dot{\boldsymbol{u}}^T\hat{\boldsymbol{w}}_{\bar{y}} = \frac{1}{r}\left((\boldsymbol{I} - \boldsymbol{u}\boldsymbol{u}^T)(\boldsymbol{w}_{\bar{y}} - \sum_i p_i\boldsymbol{w}_i)\right)^T \hat{\boldsymbol{w}}_{\bar{y}} \tag{34}$$

Since the projection matrix $(\boldsymbol{I} - \boldsymbol{u}\boldsymbol{u}^T)$ is symmetric, this simplifies to:

$$\dot{a} = \frac{1}{r}(\boldsymbol{w}_{\bar{y}} - \sum_i p_i\boldsymbol{w}_i)^T(\boldsymbol{I} - \boldsymbol{u}\boldsymbol{u}^T)\hat{\boldsymbol{w}}_{\bar{y}} = \frac{1}{r}(\boldsymbol{w}_{\bar{y}} - \sum_i p_i\boldsymbol{w}_i)^T(\hat{\boldsymbol{w}}_{\bar{y}} - \boldsymbol{u}(\boldsymbol{u}^T\hat{\boldsymbol{w}}_{\bar{y}})) \tag{35}$$

Recalling that $a = \boldsymbol{u}^T\hat{\boldsymbol{w}}_{\bar{y}}$, we get:

$$\dot{a} = \frac{1}{r}(\boldsymbol{w}_{\bar{y}} - \sum_i p_i\boldsymbol{w}_i)^T(\hat{\boldsymbol{w}}_{\bar{y}} - a\boldsymbol{u}) \tag{36}$$

To analyze the terminal phase, we note that as training progresses and classification accuracy improves, the softmax probabilities become increasingly peaked. Specifically, when the correct class logit satisfies $\ell_{\bar{y}} - \max_{i \neq \bar{y}} \ell_i \gg 1$, we have $p_{\bar{y}} = \frac{\exp(\ell_{\bar{y}})}{\exp(\ell_{\bar{y}}) + \sum_{i \neq \bar{y}} \exp(\ell_i)} \approx 1 - \sum_{i \neq \bar{y}} \exp(\ell_i - \ell_{\bar{y}}) \to 1$. Under this regime, the approximation $\sum_i p_i\boldsymbol{w}_i \approx p_{\bar{y}}\boldsymbol{w}_{\bar{y}}$ becomes increasingly accurate.

In the TPT, classification confidence is high, so $p_{\bar{y}} \to 1$ and $p_{i \neq \bar{y}} \to 0$. We can thus approximate the softmax-weighted sum of classifiers as $\sum_i p_i\boldsymbol{w}_i \approx p_{\bar{y}}\boldsymbol{w}_{\bar{y}}$. This leads to:

$$\boldsymbol{w}_{\bar{y}} - \sum_i p_i\boldsymbol{w}_i \approx \boldsymbol{w}_{\bar{y}} - p_{\bar{y}}\boldsymbol{w}_{\bar{y}} = (1 - p_{\bar{y}})\boldsymbol{w}_{\bar{y}} \tag{37}$$

Substituting this back into the expression for $\dot{a}$ yields:

$$\dot{a} \approx \frac{1}{r}\left((1 - p_{\bar{y}})\boldsymbol{w}_{\bar{y}}\right)^T (\hat{\boldsymbol{w}}_{\bar{y}} - a\boldsymbol{u}) \tag{38}$$

$$= \frac{1 - p_{\bar{y}}}{r}\left(\boldsymbol{w}_{\bar{y}}^T\hat{\boldsymbol{w}}_{\bar{y}} - a\boldsymbol{w}_{\bar{y}}^T\boldsymbol{u}\right) \tag{39}$$

Using the definitions $\hat{\boldsymbol{w}}_{\bar{y}} = \boldsymbol{w}_{\bar{y}}/\|\boldsymbol{w}_{\bar{y}}\|$ and $a = \boldsymbol{u}^T\hat{\boldsymbol{w}}_{\bar{y}} = \boldsymbol{u}^T\boldsymbol{w}_{\bar{y}}/\|\boldsymbol{w}_{\bar{y}}\|$, we have $\boldsymbol{w}_{\bar{y}}^T\hat{\boldsymbol{w}}_{\bar{y}} = \|\boldsymbol{w}_{\bar{y}}\|$ and $\boldsymbol{w}_{\bar{y}}^T\boldsymbol{u} = a\|\boldsymbol{w}_{\bar{y}}\|$. Substituting these gives:

$$\dot{a} \approx \frac{1 - p_{\bar{y}}}{r}\left(\|\boldsymbol{w}_{\bar{y}}\| - a \cdot a\|\boldsymbol{w}_{\bar{y}}\|\right) \tag{40}$$

$$= \frac{(1 - p_{\bar{y}})\|\boldsymbol{w}_{\bar{y}}\|}{r}(1 - a^2) \tag{41}$$

Since $(1 - a^2) \geq 0$ for $|a| \leq 1$ and $(1 - p_{\bar{y}}) \geq 0$, the alignment monotonically increases until $a = 1$, achieving perfect alignment with the classifier weight. $\square$

This analysis demonstrates that $L_2$ regularization not only confines representations to a bounded region but actively drives them toward geometric alignment with their corresponding classifier weights. The regularization parameter $\gamma$ controls the speed and quality of this alignment by determining the equilibrium radius $r^* = O(M_W/\gamma)$, at which representations stabilize, which influences the effective time constant of the angular dynamics. Importantly, this alignment emerges naturally from the optimization dynamics without explicit geometric constraints, revealing how strong regularization implicitly promotes convergence to the NC geometry.

### A.5 UNBOUNDED GROWTH UNDER PURE CROSS-ENTROPY MINIMIZATION

Having established the beneficial effects of strong $L_2$ regularization, we now demonstrate its necessity by analyzing the pathological behavior that emerges in its absence. This analysis serves two purposes: (1) it explains why standard neural networks without explicit regularization fail to achieve the theoretical NC predicted by UFM theory, and (2) it highlights that the confinement provided by $L_2$ regularization is not merely helpful but essential for reaching meaningful equilibria. We now demonstrate that without $L_2$ regularization, cross-entropy minimization alone leads to unbounded growth of representation norms, preventing convergence to the theoretical equilibrium predicted by UFM theory.

**Theorem A.8** (Unbounded Norm Growth without Regularization). *In the TPT with $\gamma = 0$, cross-entropy minimization drives representations to grow unboundedly: $\|\boldsymbol{z}(t)\| \to \infty$ as $t \to \infty$.*

*Proof.* For pure cross-entropy loss with $\gamma = 0$, the gradient is:

$$\nabla_{\boldsymbol{z}}\mathcal{L}_{\text{CE}}(\boldsymbol{z}) = \sum_{i=1}^K p_i(\boldsymbol{z})\boldsymbol{w}_i - \boldsymbol{w}_{\bar{y}} \tag{42}$$

Consider the rate of change of the squared norm:

$$\frac{d}{dt}\|\boldsymbol{z}\|^2 = 2\boldsymbol{z}^T\dot{\boldsymbol{z}} = -2\boldsymbol{z}^T\nabla_{\boldsymbol{z}}\mathcal{L}_{\text{CE}}(\boldsymbol{z}) \tag{43}$$

$$= 2\boldsymbol{z}^T\left(\boldsymbol{w}_{\bar{y}} - \sum_{i=1}^K p_i(\boldsymbol{z})\boldsymbol{w}_i\right) \tag{44}$$

$$= 2\left(\ell_{\bar{y}} - \sum_{i=1}^K p_i(\boldsymbol{z})\ell_i\right) \tag{45}$$

where $\ell_i = \boldsymbol{z}^T\boldsymbol{w}_i$ are the logits. The term $\ell_{\bar{y}} - \sum_i p_i\ell_i$ represents the difference between the correct class logit and the expected logit over the softmax distribution. This quantity is equivalent to $\sum_i p_i(\ell_{\bar{y}} - \ell_i)$ and is strictly positive as long as perfect classification ($p_{\bar{y}} = 1$) has not been achieved, since $\ell_{\bar{y}}$ will be greater than other logits $\ell_i$ for which $p_i > 0$. Therefore, $\frac{d}{dt}\|\boldsymbol{z}\|^2 > 0$, indicating that under gradient flow in the terminal phase, the steepest descent direction for CE loss has an outward radial component, causing representation norms to grow continuously.

Crucially, cross-entropy can be decreased by simply scaling up $\boldsymbol{z}$ without requiring alignment. If $\boldsymbol{z}$ makes an angle $\theta$ with $\boldsymbol{w}_{\bar{y}}$ where $\cos\theta < 1$, scaling $\boldsymbol{z} \mapsto \alpha\boldsymbol{z}$ with $\alpha > 1$ yields:

$$\mathcal{L}_{\text{CE}}(\alpha\boldsymbol{z}) = -\log\frac{\exp(\alpha\ell_{\bar{y}})}{\sum_i \exp(\alpha\ell_i)} \to 0 \text{ as } \alpha \to \infty \tag{46}$$

provided $\ell_{\bar{y}} > \max_{i \neq \bar{y}} \ell_i$, which only requires $\cos\theta > \cos\theta_{\text{critical}}$ for some critical angle $\theta_{\text{critical}} < \pi/2$. $\qquad\square$

**Corollary A.9** (Misalignment Compatible with Loss Decrease). *Even with slight misalignment between $\boldsymbol{z}$ and $\boldsymbol{w}_{\bar{y}}$, cross-entropy loss can decrease through norm growth alone. Specifically, if:*

$$\boldsymbol{z}^T \boldsymbol{w}_{\bar{y}} > \max_{i \neq \bar{y}} \boldsymbol{z}^T \boldsymbol{w}_i \tag{47}$$

*then increasing $\|\boldsymbol{z}\|$ while maintaining fixed direction decreases $\mathcal{L}_{CE}(\boldsymbol{z})$.*

This unbounded growth has critical consequences:

**Proposition A.10** (Gradient Saturation and Stalled Convergence). *As $\|\boldsymbol{z}\| \to \infty$ without regularization*

 1. *The gradient norm vanishes: $\|\nabla_{\boldsymbol{z}} \mathcal{L}_{CE}(\boldsymbol{z})\| \to 0$*

 2. *The loss plateaus: $\mathcal{L}_{CE}(\boldsymbol{z}) \to 0$ at rate $O(e^{-\|\boldsymbol{z}\|})$*

 3. *Convergence to any finite equilibrium point becomes impossible*

*Proof.* As $\|\boldsymbol{z}\| \to \infty$ with $\ell_{\bar{y}} > \max_{i \neq \bar{y}} \ell_i$:

$$p_{\bar{y}}(\boldsymbol{z}) = \frac{\exp(\ell_{\bar{y}})}{\sum_i \exp(\ell_i)} \to 1 \tag{48}$$

The gradient becomes:

$$\|\nabla_{\boldsymbol{z}} \mathcal{L}_{\text{CE}}(\boldsymbol{z})\| = \left\| \sum_i p_i \boldsymbol{w}_i - \boldsymbol{w}_{\bar{y}} \right\| = (1 - p_{\bar{y}}) \|\boldsymbol{w}_{\bar{y}} - \bar{\boldsymbol{w}}\| \to 0 \tag{49}$$

With vanishing gradients, the dynamics essentially halt, preventing convergence to any finite equilibrium point. The representations are trapped in a regime of infinite growth with diminishing returns in loss reduction. $\qquad\square$

This analysis reveals that pure cross-entropy optimization, despite achieving low loss values, fails to reach the structured equilibria predicted by theory. The unbounded norm growth and gradient saturation prevent the formation of the NC geometry. Strong $L_2$ regularization with large $\gamma$ is therefore essential to constrain representations to a bounded region where meaningful equilibria can be reached.

## A.6 GLOBAL LIPSCHITZ CONTINUITY

The confinement effects of strong $L_2$ regularization lead to a crucial property: global Lipschitz continuity of the network.

**Theorem A.11** (Lipschitz Bound). *For a neural network with $L_2$-regularized penultimate representations trained with any optimizer satisfying Assumption A.2, after sufficient training with learning rate $\eta < \frac{1}{\gamma\bar{\lambda}}$, the expected network output difference satisfies:*

$$\mathbb{E}[\|\boldsymbol{f}(\boldsymbol{x}_1) - \boldsymbol{f}(\boldsymbol{x}_2)\|] \le L(\gamma) \tag{50}$$

*where the global Lipschitz constant is:*

$$L(\gamma) = 2\sqrt{K} M_W \left( \frac{M_W}{\gamma} + \sqrt{\frac{d\eta\bar{\lambda}\sigma^2}{4\gamma(1 - \eta\gamma\bar{\lambda})}} \right) \tag{51}$$

*This bound is independent of the input distance $\|\boldsymbol{x}_1 - \boldsymbol{x}_2\|$.*

*Proof.* For any two inputs $\boldsymbol{x}_1, \boldsymbol{x}_2$, their penultimate-layer representations are $\boldsymbol{z}_1, \boldsymbol{z}_2$. The distance between them is bounded by the triangle inequality: $\|\boldsymbol{z}_1 - \boldsymbol{z}_2\| \leq \|\boldsymbol{z}_1\| + \|\boldsymbol{z}_2\|$.

From Theorem A.4, any equilibrium point $\boldsymbol{z}^*$ satisfies $\|\boldsymbol{z}^*\| \leq M_W / \gamma$. From the Hessian analysis, the steady-state deviation from equilibrium satisfies:

$$\mathbb{E}[\|\boldsymbol{d}_\infty\|^2] = \frac{d\eta\bar{\lambda}\sigma^2}{4\gamma(1 - \eta\gamma\bar{\lambda})} \tag{52}$$

For the expected norm, we use the fact that for any random vector, $\mathbb{E}[\|\boldsymbol{v}\|] \leq \sqrt{\mathbb{E}[\|\boldsymbol{v}\|^2]}$ by Jensen's inequality:

$$\mathbb{E}[\|\boldsymbol{d}_\infty\|] \leq \sqrt{\mathbb{E}[\|\boldsymbol{d}_\infty\|^2]} = \sqrt{\frac{d\eta\bar{\lambda}\sigma^2}{4\gamma(1 - \eta\gamma\bar{\lambda})}} \tag{53}$$

Therefore, the expected norm of any representation satisfies:

$$\mathbb{E}[\|\boldsymbol{z}\|] \leq \|\boldsymbol{z}^*\| + \mathbb{E}[\|\boldsymbol{d}\|] \leq \frac{M_W}{\gamma} + \sqrt{\frac{d\eta\bar{\lambda}\sigma^2}{4\gamma(1 - \eta\gamma\bar{\lambda})}} \tag{54}$$

Since this bound holds in expectation for any representation, we can bound the expected distance between any two representations:

$$\mathbb{E}[\|\boldsymbol{z}_1 - \boldsymbol{z}_2\|] \leq \mathbb{E}[\|\boldsymbol{z}_1\|] + \mathbb{E}[\|\boldsymbol{z}_2\|] \leq 2\left(\frac{M_W}{\gamma} + \sqrt{\frac{d\eta\bar{\lambda}\sigma^2}{4\gamma(1 - \eta\gamma\bar{\lambda})}}\right) \tag{55}$$

The network output difference is $\boldsymbol{f}(\boldsymbol{x}_1) - \boldsymbol{f}(\boldsymbol{x}_2) = \boldsymbol{W}(\boldsymbol{z}_1 - \boldsymbol{z}_2)$. Using the spectral norm bound $\|\boldsymbol{W}\|_2 \leq \sqrt{K}M_W$ from Lemma A.5:

$$\|\boldsymbol{f}(\boldsymbol{x}_1) - \boldsymbol{f}(\boldsymbol{x}_2)\| \leq \|\boldsymbol{W}\|_2\|\boldsymbol{z}_1 - \boldsymbol{z}_2\| \leq \sqrt{K}M_W\|\boldsymbol{z}_1 - \boldsymbol{z}_2\| \tag{56}$$

Therefore, taking expectations over the steady-state distribution:

$$\mathbb{E}[\|\boldsymbol{f}(\boldsymbol{x}_1) - \boldsymbol{f}(\boldsymbol{x}_2)\|] \leq \sqrt{K}M_W \cdot \mathbb{E}[\|\boldsymbol{z}_1 - \boldsymbol{z}_2\|] \leq L(\gamma) \tag{57}$$

which establishes the expected Lipschitz bound. $\square$

The structure of $L(\gamma)$ reveals that both terms decrease as $\gamma$ increases:

$$L(\gamma) = 2\sqrt{K}M_W \underbrace{\left(\frac{M_W}{\gamma}\right)}_{\text{equilibrium term}} + 2\sqrt{K}M_W \underbrace{\left(\sqrt{\frac{d\eta\bar{\lambda}\sigma^2}{4\gamma(1 - \eta\gamma\bar{\lambda})}}\right)}_{\text{stochastic fluctuation term}} \tag{58}$$

The first term decreases as $O(1/\gamma)$ while the second decreases as $O(1/\sqrt{\gamma})$ for small learning rates where $\eta\gamma\bar{\lambda} \ll 1$. Therefore, increasing the regularization strength $\gamma$ monotonically improves the network's Lipschitz constant, with the dominant improvement coming from the $O(1/\gamma)$ reduction in the equilibrium bound.

**Remark A.12** (High-Probability Bound). *While we establish the Lipschitz bound in expectation, a high-probability bound can be obtained using concentration inequalities. For instance, by Markov's inequality, for any $\delta > 0$:*

$$\Pr[\|\boldsymbol{f}(\boldsymbol{x}_1) - \boldsymbol{f}(\boldsymbol{x}_2)\| > L(\gamma)/\delta] \leq \delta \tag{59}$$

## A.7 MAIN RESULT

**Theorem A.13** (Latent Point Collapse under Strong $L_2$ Regularization). *For a neural network with bounded classifier weights $\|\boldsymbol{w}_i\| \leq M_W$ and $L_2$-regularized penultimate representations with parameter $\gamma$, trained with an optimizer satisfying Assumption A.2 and learning rate $\eta < \frac{1}{\gamma\bar{\lambda}}$:*

1. **Confinement:** *All equilibrium representations satisfy* $\|\boldsymbol{z}^*\| \leq \frac{M_W}{\gamma}$

2. **Collapse:** *Under stochastic optimization, intra-class representations concentrate within radius:*

$$r_{collapse} = \sqrt{\mathbb{E}[\|\boldsymbol{d}_\infty\|^2]} = \frac{\sigma}{2}\sqrt{\frac{d\eta\bar{\lambda}}{\gamma(1 - \eta\gamma\bar{\lambda})}} = O\left(\sqrt{\frac{\sigma^2 d\eta}{\gamma}}\right) \tag{60}$$

*where the asymptotic form holds for small learning rates* $\eta\gamma\bar{\lambda} \ll 1$.

3. **Alignment:** *In the terminal phase of training where* $p_{\bar{y}} \to 1$, *representations align with their corresponding classifier weights:* $\lim_{t\to\infty} \frac{\boldsymbol{z}(t)}{\|\boldsymbol{z}(t)\|} = \frac{\boldsymbol{w}_{\bar{y}}}{\|\boldsymbol{w}_{\bar{y}}\|}$

4. **Global Lipschitz Continuity:** *For any inputs* $\boldsymbol{x}_1, \boldsymbol{x}_2$, *in expectation:*

$$\mathbb{E}[\|\boldsymbol{f}(\boldsymbol{x}_1) - \boldsymbol{f}(\boldsymbol{x}_2)\|] \leq \frac{2\sqrt{K}M_W^2}{\gamma} + O\left(\sqrt{\frac{\sigma^2 d\eta}{\gamma}}\right) \tag{61}$$

*This bound is* **independent of input distance** $\|\boldsymbol{x}_1 - \boldsymbol{x}_2\|$.

5. **Necessity of Regularization:** *Without* $L_2$ *regularization* ($\gamma = 0$), *cross-entropy minimization alone causes unbounded norm growth* ($\|\boldsymbol{z}\| \to \infty$), *gradient saturation, and failure to reach theoretical equilibria.*

## A.8 SUMMARY

Strong $L_2$ regularization with parameter $\gamma$ is both necessary and sufficient to achieve LPC while ensuring global Lipschitz continuity. Under the assumption of a sufficiently small learning rate $\eta < \frac{1}{\gamma\bar{\lambda}}$ for convergence, the regularization parameter provides a unified mechanism that:

1. **Prevents representation explosion:** Creates an inward force proportional to $2\gamma\boldsymbol{z}$ that counteracts the outward bias of gradient descent in the terminal phase. Without this regularization, gradient-based optimization of cross-entropy loss leads to unbounded norm growth, as the steepest descent direction has an outward radial component whenever $p_{\bar{y}} < 1$ (Theorem A.8).

2. **Induces intra-class collapse:** For $\gamma > \frac{KM_W^2}{2}$, establishes a strongly convex loss landscape. Under stochastic optimization, representations concentrate around class-specific equilibria with collapse radius $r_{\text{collapse}} = O(\sqrt{\sigma^2 d\eta/\gamma})$, where the primary control is through $\gamma$. The steady-state variance $\mathbb{E}[\|\boldsymbol{d}_\infty\|^2] = \frac{d\eta\bar{\lambda}\sigma^2}{4\gamma(1-\eta\gamma\bar{\lambda})}$ decreases monotonically as $\gamma$ increases.

3. **Drives weight alignment:** In the terminal phase where $p_{\bar{y}} \to 1$, the angular dynamics decouple from the radial dynamics, causing representations to align progressively with their corresponding classifier weights. The alignment rate $(1 - a^2)(1 - p_{\bar{y}})\|\boldsymbol{w}_{\bar{y}}\|/r$ depends on the current alignment $a$ and the equilibrium radius $r = O(M_W/\gamma)$.

4. **Ensures global Lipschitz continuity:** By confining representations to a bounded region, yields an expected global Lipschitz constant:

$$L(\gamma) = 2\sqrt{K}M_W\left(\frac{M_W}{\gamma} + O\left(\sqrt{\frac{\sigma^2 d\eta}{\gamma}}\right)\right) \tag{62}$$

Both terms decrease with increasing $\gamma$, with the dominant $O(1/\gamma)$ term providing the primary improvement. This bound is independent of input distance $\|\boldsymbol{x}_1 - \boldsymbol{x}_2\|$, establishing uniform stability across the input space.

5. **Enables convergence to global optima:** For $\gamma > \frac{KM_W^2}{2}$, creates a single-basin landscape with unique global minimum. Combined with appropriate learning rate $\eta < \frac{1}{\gamma\bar{\lambda}}$, guarantees convergence to a neighborhood of the global optimum, with neighborhood radius controlled by the regularization strength.

Without large $\gamma$, cross-entropy optimization alone produces unbounded representation growth without achieving the structured geometry predicted by UFM theory. In the terminal phase, gradient descent follows a path with an outward radial component, leading to gradient saturation as $\|\nabla_{\boldsymbol{z}} \mathcal{L}_{\mathrm{CE}}\| \to 0$ and preventing convergence to finite equilibria. Strong $L_2$ regularization resolves this pathology by providing a countervailing inward force, transforming an ill-posed optimization problem into a well-conditioned one with provable convergence guarantees and explicit control over both geometric structure and robustness properties.

## B    Information Bottleneck in Deterministic DNN Classifiers

The connection between margin-based approaches and robust generalization can also be understood through the framework of the information bottleneck (IB) principle (89; 90). The IB principle suggests that DNNs seek compact yet sufficiently informative latent representations by minimizing the mutual information between inputs and latent representations, while preserving information relevant for prediction. Empirically, it has been shown that IB improves network performance (91), and theoretical work provides rigorous arguments for IB's role in controlling generalization errors (92). In practice, DNN training reveals two distinct phases: an empirical risk minimization phase, where the network primarily fits the data, followed by a compression phase, where the network constructs more compact embeddings layer by layer (93). This compression aligns with margin maximization and NC, suggesting that the pursuit of efficient representations manifests in both information-theoretic and geometric properties.

The emergence of LPC creates an IB in the latent space, connecting this phenomenon to IB optimization in DNNs (90; 94; 95; 96; 97; 98). Unlike many IB methods, which rely on variational approximations or noise injection, LPC implements a deterministic form of compression through a strong $L_2$ penalty on the features themselves, effectively shrinking their distribution and lowering their entropy.

The proposed loss function induces the collapse of all same-class latent representations into a single point, which can also be posed as a method to create an IB in the penultimate-layer. The optimization of the IB Lagrangian aims to maximize the following objective:

$$\mathcal{L}_{IB} = I(\mathbf{z}; \mathbf{y}) - \beta I(\mathbf{z}; \mathbf{x}), \tag{63}$$

where $I(\mathbf{z}; \mathbf{y})$ denotes the mutual information between the latent representation $\mathbf{z}$ and the labels y and $I(\mathbf{z}; \mathbf{x})$ represents the mutual information between $\mathbf{z}$ and the input data $\mathbf{x}$. The parameter $\beta$ controls the trade-off between compression and predictive accuracy. In App. B, we demonstrate that minimizing this quantity in deterministic DNN classification is equivalent to minimizing:

$$\mathcal{L}_{IB} = \mathcal{L}_{\mathrm{CE}}(\boldsymbol{f}(\boldsymbol{x}), \bar{y}) + \beta H(\mathbf{z}), \tag{64}$$

where $H(\mathbf{z})$ is the entropy associated with the latent distribution $\mathbf{z}$ and $\mathcal{L}_{\mathrm{CE}}(\boldsymbol{f}(\boldsymbol{x}), \bar{y})$ is the cross-entropy loss function. During training, the cross-entropy loss is directly minimized, while the entropy $H(\mathbf{z})$ is indirectly minimized by the collapse of all same-class latent representations into a single point. To understand how LPC effectively minimizes the entropy of the probability distributions generating latent representations $\boldsymbol{z}$, we approximate the differential entropy with a discrete Shannon entropy and take the limit for an infinitesimally small quantization: $H_\Delta = -\sum_i p_i \log p_i$. As a result of the collapse of all same-class latent representations into a single point, all elements of a specific class are confined to a unique bin, even with very small bin size. For $K$ classes with equal elements per class, the entropy reduces to: $H_\Delta = -\log \frac{1}{K}$. This represents the minimum possible entropy value that still permits discrimination among classes. If the latent representations do not collapse into a single point, the distribution will spread across multiple bins, resulting in higher entropy.

The IB objective can be formulated as an optimization problem (89), aiming to maximize the following function:

$$\mathcal{L}_{IB} = I(\mathbf{z}; y) - \beta I(\mathbf{z}; \mathbf{x}),$$

where $I(\mathbf{z}; y)$ denotes the mutual information between the latent representation $\mathbf{z}$ and the labels $y$, while $I(\mathbf{z}; \mathbf{x})$ represents the mutual information between $\mathbf{z}$ and the input data $\mathbf{x}$. The parameter $\beta$ controls the trade-off between compression and predictive accuracy. Our goal is to maximize the

Table 3: All values in the table represent the means and standard deviations obtained from different experiments. The table shows the estimated entropy ($H$) on the testing set using the Kozachenko-Leonenko method (k=20), divided by the penultimate-layer dimension.

| | CIFAR-10 | | CIFAR-100 | IMAGENET |
|---|---|---|---|---|
| MODEL | ENTROPY | | ENTROPY | ENTROPY |
| LPC | $-3.96$ | $\pm 0.31$ | $-2.106 \pm 0.119$ | $1.323 \pm 0.004$ |
| LPC-WIDE | $-3.4$ | $\pm 0.34$ | $-1.914 \pm 0.15$ | $-$ |
| LPC-NARROW | $-4.13$ | $\pm 0.34$ | $-2.319 \pm 0.04$ | $-$ |
| LPC-SCL | $-3.27$ | $\pm 0.11$ | $-1.675 \pm 0.089$ | $-$ |
| LPC-NOPEN | $-1.77$ | $\pm 0.33$ | $5.559 \pm 0.054$ | $-$ |
| LINPEN | $0.58$ | $\pm 0.26$ | $1.257 \pm 0.022$ | $-$ |
| NONLINPEN | $0.46$ | $\pm 0.02$ | $1.208 \pm 0.006$ | $-$ |
| SCL | $0.24$ | $\pm 0.4$ | $5.123 \pm 0.097$ | $-$ |
| ARCFACE | $-0.06$ | $\pm 0.13$ | $5.055 \pm 0.02$ | $-$ |
| NOPEN | $0.88$ | $\pm 0.05$ | $5.566 \pm 0.029$ | $5.916 \pm 0.002$ |
| NOPENWD | $0.59$ | $\pm 0.02$ | $5.529 \pm 0.017$ | $-$ |

mutual information between the latent representations and the labels, $I(\mathbf{z}; y)$. This mutual information can be expressed in terms of entropy:

$$I(\mathbf{z}; y) = H(y) - H(y|\mathbf{z}),$$

where $H(y)$ is the entropy of the labels and $H(y|\mathbf{z})$ is the conditional entropy of the labels given the latent representations. Since $H(y)$ is constant with respect to the model parameters (as it depends solely on the distribution of the labels), maximizing $I(\mathbf{z}; y)$ is equivalent to minimizing the conditional entropy $H(y|\mathbf{z})$:

$$\max I(\mathbf{z}; y) \quad \Leftrightarrow \quad \min H(y|\mathbf{z}).$$

The conditional entropy $H(y|\mathbf{z})$ can be estimated empirically using the dataset. Assuming that the data points $(\mathbf{x}^{(n)}, y^{(n)})$ are sampled from the joint distribution $p(\mathbf{x}, y)$ and that $\mathbf{z}^{(n)} = f(\mathbf{x}^{(n)})$, we approximate $H(y|\mathbf{z})$ as:

$$H(y|\mathbf{z}) \approx -\frac{1}{N} \sum_{n=1}^{N} \sum_{k=1}^{K} p(y_k|\mathbf{z}^{(n)}) \log p(y_k|\mathbf{z}^{(n)}),$$

where $K$ is the number of classes and $p(y_k|\mathbf{z}^{(n)})$ is the probability of label $y_k$ given latent representation $\mathbf{z}^{(n)}$. In practice, since we have the true labels $y^{(n)}$, this simplifies to:

$$H(y|\mathbf{z}) \approx -\frac{1}{N} \sum_{n=1}^{N} \log p(y^{(n)}|\mathbf{z}^{(n)}).$$

This expression corresponds to the cross-entropy loss commonly used in training classifiers. In a DNN classifier, the probability $p(y|\mathbf{z})$ is modeled using the softmax function applied to the output logits:

$$p(y_k|\mathbf{z}) = \frac{\exp\left((\mathbf{W}\mathbf{z} + \mathbf{b})_k\right)}{\sum_{i=1}^{K} \exp\left((\mathbf{W}\mathbf{z} + \mathbf{b})_i\right)},$$

where $\mathbf{W}$ and $\mathbf{b}$ are the weights and biases of the final layer, and $(\mathbf{W}\mathbf{z} + \mathbf{b})_k$ denotes the logit corresponding to class $y_k$. By minimizing $H(y|\mathbf{z})$, we encourage the model to produce latent representations that are informative about the labels, aligning with the objective of accurate classification.

Table 4: Summary of the features implemented in all architectures used in our ablation study. *Lin. Pen* refers to the inclusion or exclusion of a linear penultimate-layer. *Nodes Add. Layer* feature indicates the presence of an additional layer between the backbone and the classification layer. If this layer is present, its dimensionality is categorized as one of three possible values: wide, intermediate, or narrow. The exact dimensionality for these categories is a hyperparameter that varies across different datasets. *Loss* indicates the type of loss function utilized during training.

| Model | Lin. Pen. | Nodes Add. Layer | Loss |
|---|---|---|---|
| LPC | ✓ | Intermediate | $CE + L_2$ |
| LPC-Wide | ✓ | Wide | $CE + L_2$ |
| LPC-Narrow | ✓ | Narrow | $CE + L_2$ |
| LPC-SCL | ✓ | Intermediate | $CE + L_2 + SCL$ |
| LPC-NoPen | ✗ | ✗ | $CE + L_2$ |
| LinPen | ✓ | Intermediate | $CE$ |
| NonlinPen | ✗ | Intermediate | $CE$ |
| SCL | ✗ | ✗ | $CE + SCL$ |
| ArcFace | ✗ | ✗ | $ArcFace$ |
| NoPenWD | ✗ | ✗ | $CE$ |
| NoPen | ✗ | ✗ | $CE$ |

The second term in the IB objective, $I(\mathbf{z}; \mathbf{x})$, quantifies the mutual information between the latent representations and the inputs. To achieve compression, we aim to minimize this term. Expressing $I(\mathbf{z}; \mathbf{x})$ in terms of entropy:

$$I(\mathbf{z}; \mathbf{x}) = H(\mathbf{z}) - H(\mathbf{z}|\mathbf{x}).$$

In the case of deterministic mappings where $\mathbf{z} = f(\mathbf{x})$, the differential conditional entropy $H(\mathbf{z}|\mathbf{x})$ is ill-defined, therefore we focus solely on minimizing $H(\mathbf{z})$ as explained in the InfoMax seminal paper (99).

$$\min I(\mathbf{z}; \mathbf{x}) \quad \Leftrightarrow \quad \min H(\mathbf{z}).$$

To empirically validate our theoretical analysis, we conducted extensive experiments as detailed in Section 3. Table 3 presents the estimated entropy values using the Kozachenko-Leonenko method for various models. The results clearly demonstrate that LPC-based models exhibit significantly lower entropy values compared to their non-penalized counterparts. This significant reduction in entropy occurs because LPC confines all same-class latent representations to a single point, effectively minimizing $H(\mathbf{z})$ in the IB objective. As we demonstrated, when all elements of a specific class collapse to a unique location, even with infinitesimally small quantization, the entropy reduces to the minimum possible value that still permits discrimination among classes: $H_\Delta = -\log \frac{1}{K}$ for $K$ classes. This experimental evidence confirms that LPC serves as an effective method to create an information bottleneck in the penultimate-layer, achieving substantial compression while maintaining discriminative capabilities necessary for classification.

## C  Training and Architecture Details.

Our ablation study systematically evaluates each component of the proposed method. All architectures employ a shared backbone network that produces the latent representation $\boldsymbol{h}(\boldsymbol{x})$, while differing in their approach to final classification.

We denote the architecture with $L_2$ regularization applied to a linear penultimate-layer of intermediate dimensionality as LPC, with its lower-dimensional and higher-dimensional variants designated as LPC-Narrow and LPC-Wide, respectively. These three variants examine the effect of penultimate-layer dimensionality on LPC formation and network performance. To isolate the contribution of $L_2$ regularization, we include LinPen and NonlinPen controls—linear and non-linear penultimate-layers matching LPC's dimensionality but trained exclusively with cross-entropy loss.

Table 5: Average training time per epoch (in minutes) for different models across datasets. All values represent means and standard deviations obtained from different experiments. Experiments were conducted using 1 NVIDIA A100 GPU for CIFAR-10 and CIFAR-100, and 2 NVIDIA A100 GPUs for ImageNet.

| MODEL | CIFAR-10 | CIFAR-100 | IMAGENET |
|---|---|---|---|
| LPC | $0.103 \pm 0.000$ | $0.325 \pm 0.001$ | $58.86 \pm 0.44$ |
| LPC-WIDE | $0.104 \pm 0.000$ | $0.327 \pm 0.000$ | – |
| LPC-NARROW | $0.103 \pm 0.000$ | $0.325 \pm 0.000$ | – |
| LPC-SCL | $0.106 \pm 0.000$ | $0.327 \pm 0.001$ | – |
| LPC-NOPEN | $0.103 \pm 0.000$ | $0.332 \pm 0.001$ | – |
| LINPEN | $0.104 \pm 0.000$ | $0.328 \pm 0.001$ | – |
| NONLINPEN | $0.103 \pm 0.000$ | $0.326 \pm 0.001$ | – |
| SCL | $0.105 \pm 0.000$ | $0.335 \pm 0.001$ | – |
| ARCFACE | $0.107 \pm 0.001$ | $0.336 \pm 0.001$ | – |
| NOPEN | $0.104 \pm 0.000$ | $0.334 \pm 0.001$ | $33.81 \pm 0.28$ |
| NOPENWD | $0.103 \pm 0.000$ | $0.334 \pm 0.000$ | – |

The LPC-NOPEN model tests whether the penultimate-layer structure is necessary by applying $L_2$ regularization directly to the backbone output $\boldsymbol{h}(\boldsymbol{x})$ and, unlike other models, excludes an intermediate layer. The NOPEN model serves as our baseline, performing linear classification directly on $\boldsymbol{h}(\boldsymbol{x})$ using only cross-entropy loss. To demonstrate that LPC benefits cannot be attributed solely to stronger weight decay regularization, we include NOPENWD, which employs the same architecture as NOPEN but with substantially increased weight decay.

We compare against SCL and ARCFACE baselines, which implement their respective loss functions on the NOPEN baseline architecture. The LPC-SCL architecture combines $L_2$ regularization on an intermediate penultimate linear layer with SupCon applied to the backbone's latent representations, testing compatibility with other metric learning approaches. Note that in LPC-SCL, these losses operate on distinct network layers.

All experiments were conducted on CIFAR-10, CIFAR-100 (81), and ImageNet-1K (82) datasets. To generate latent representations $\boldsymbol{h}(\boldsymbol{x})$, we employed ResNet (83) backbones tailored to dataset complexity: ResNet-18 for CIFAR-10, ResNet-50 for CIFAR-100, and WideResNet-50 (84) for ImageNet. All architectures incorporated batch normalization and Swish activation functions (100) throughout.

Architecture configurations varied by method: LPC, LPC-SCL, LINPEN, and NONLINPEN included a 64-dimensional fully connected penultimate-layer. LPC-WIDE expanded this to 128 dimensions, while LPC-NARROW reduced it to 32 dimensions for CIFAR datasets. For ImageNet, LPC employed 128 penultimate dimensions. The LPC-NOPEN, NOPEN, NOPENWD, SCL, and ARCFACE architectures omitted the penultimate-layer entirely (see Table 4).

We trained models using AdamW (101; 102) with default PyTorch parameters. All experiments used a batch size of 128 for CIFAR datasets and 64 for ImageNet. Weight decay was set to $5 \times 10^{-4}$ for CIFAR datasets and $1 \times 10^{-7}$ for ImageNet. In the architecture NOPENWD weight decay had a larger value set to $1 \times 10^{-1}$. Data augmentation comprised random horizontal flips and random crops (padding=4) for CIFAR datasets, while ImageNet used only random cropping to 224×224 pixels. We evaluated four learning rates from a geometric sequence $(10^{-4}, 2 \times 10^{-4}, 4 \times 10^{-4}, 8 \times 10^{-4})$ for CIFAR experiments and three rates $(5 \times 10^{-5}, 1 \times 10^{-4}, 1 \times 10^{-3})$ for ImageNet, selecting the best-performing configuration based on final test accuracy. For ARCFACE, we initialized the classifier bias $b$ to zero following (74).

Training protocols differed by dataset: CIFAR models trained for 1,000 epochs with cosine annealing from epoch 200 (reducing learning rate to $1 \times 10^{-8}$), excluding the final classifier layer. ImageNet models trained for 200 epochs with annealing from epoch 70 (minimum $1 \times 10^{-7}$). Extended training ensured CIFAR experiments operated predominantly in the terminal phase of training (TPT), defined as achieving >99.9% training accuracy (11). ImageNet experiments did not reach TPT. We note that

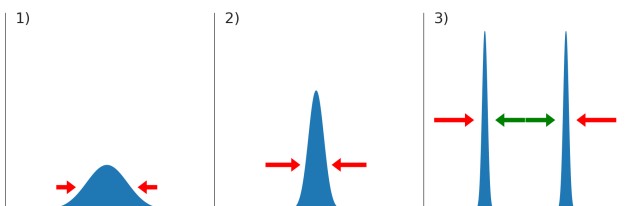

Figure 4: Graphical illustration of the dynamics leading to the emergence of binary encoding. The three images give a qualitative representation of the outcome of a training where the scalar $\gamma$ is progressively increased - from left to right - during training. Plots in the images represent histograms of the latent representations in a specific node of the linear penultimate-layer. In the first image, the relatively low value of $\gamma$ constrains all values close to the origin, but the volume is still large enough for the network to differentiate between different classes in the volume. As the magnitude of $\gamma$ is increased, all latent values are drawn closer to the origin, as depicted in the second image, and it becomes increasingly more difficult for the network to discriminate between elements of different classes. Consequently, the network is forced to find, through numerical optimization, a more stable solution by placing all elements belonging to the same class in the neighborhood of one of two points. These points are positioned opposite to each other with respect to the origin, as illustrated in the third image. The red (green) arrow represents the net effects of the binary encoding (cross-entropy) loss.

learning rate decay begins when the $L_2$ coefficient $\gamma$ is already quite large (though still increasing), which we found provides additional training stability during the later stages of optimization.

For $L_2$-regularized architectures, we initialized the coefficient $\gamma = 10^{-4}$ and increased it geometrically: $\gamma \leftarrow \gamma \cdot \gamma_{\text{step}}$ every 5 epochs (CIFAR) or 10 epochs (ImageNet). We set $\gamma_{\text{step}} = 2$ until $\gamma = 10^3$, then $\gamma_{\text{step}} = 1.25$ until reaching $\gamma_{\text{max}} = 10^6$, where it remained constant.

The supervised contrastive loss (SCL) (69) was defined as:

$$\mathcal{L}^{\text{SCL}} = -\frac{1}{N} \sum_{i=1}^{N} \frac{1}{|P(i)|} \sum_{p \in P(i)} \log \frac{\exp(s_{i,p}/\tau)}{\sum_{j \neq i} \exp(s_{i,j}/\tau)}, \tag{65}$$

where $N$ denotes batch size, $P(i)$ represents positive samples for instance $i$, $s_{i,j}$ is the cosine similarity between samples $i$ and $j$, and $\tau = 0.05$ is the temperature parameter. We jointly optimized $\mathcal{L}^{\text{SCL}}$ with cross-entropy loss for classification.

For ArcFace (74), we applied an angular margin to scaled cosine similarities before computing cross-entropy:

$$\mathcal{L}^{\text{ArcFace}} = \frac{1}{N} \sum_{i=1}^{N} \text{CE}\left(\text{Softmax}[s \cdot \cos(\theta_i + m)], y_i\right), \tag{66}$$

where the angular margin $m$ increased from 0.1 to 0.5 and scale factor $s$ from 16 to 64 during training.

All experiments were repeated five times with different random seeds. Results report mean ± standard deviation across trials.

Table 5 reports the computational cost of each architecture. Surprisingly, LPC exhibits comparable or slightly faster training times than baseline methods on CIFAR datasets, despite the additional penultimate-layer and $L_2$ regularization computations. However, on ImageNet, LPC requires substantially longer training time per epoch compared to NoPEN, reflecting the increased computational overhead of the iteratively scaled $L_2$ penalty on larger-scale datasets with higher-dimensional features.

## D  BINARITY HYPOTHESIS

Our assumption is that each dimension on the penultimate latent representation can assume approximately only one of two values, as illustrated in Fig. 4. In order to verify this assumption, we fit a Gaussian mixture model (GMM) with 2 modes on each set of latent representations

Table 6: Score $\overline{\ell}$ and relative distance of the two distances $\overline{\mu}$ over all penultimate nodes in the training set across all experiments at the last epoch. Average and min values are shown. The coefficient of variation measures the standard deviation of the norm of latent representations normalized by the mean.

| | DATASET: CIFAR-10 | | | | |
| MODEL | SCORE | MIN SCORE | PEAKS DIST | MIN PEAKS DIST | COEFF. OF VAR. |
|---|---|---|---|---|---|
| LPC | $0.37 \pm 0.26$ | $-0.06 \pm 0.34$ | $24.85 \pm 6.70$ | $15.82 \pm 5.34$ | $0.007 \pm 0.003$ |
| LPC-WIDE | $0.05 \pm 0.18$ | $-0.57 \pm 0.30$ | $18.07 \pm 3.62$ | $9.25 \pm 2.88$ | $0.012 \pm 0.003$ |
| LPC-NARROW | $0.42 \pm 0.32$ | $-0.10 \pm 0.42$ | $26.10 \pm 8.76$ | $15.25 \pm 6.14$ | $0.008 \pm 0.004$ |
| LINPEN | $-1.36 \pm 0.05$ | $-1.42 \pm 0.00$ | $1.93 \pm 0.41$ | $0.40 \pm 0.38$ | $0.339 \pm 0.030$ |
| | DATASET: CIFAR-100 | | | | |
| MODEL | SCORE | MIN SCORE | PEAKS DIST | MIN PEAKS DIST | COEFF. OF VAR. |
| LPC | $0.65 \pm 0.16$ | $0.47 \pm 0.15$ | $32.13 \pm 5.84$ | $26.07 \pm 4.05$ | $0.005 \pm 0.000$ |
| LPC-WIDE | $0.52 \pm 0.23$ | $0.07 \pm 0.31$ | $28.30 \pm 6.63$ | $20.16 \pm 8.15$ | $0.021 \pm 0.019$ |
| LPC-NARROW | $0.82 \pm 0.02$ | $0.63 \pm 0.05$ | $37.40 \pm 0.79$ | $30.89 \pm 1.26$ | $0.005 \pm 0.000$ |
| LINPEN | $-1.42 \pm 0.00$ | $-1.42 \pm 0.00$ | $1.26 \pm 0.05$ | $0.76 \pm 0.20$ | $0.258 \pm 0.013$ |
| | DATASET: IMAGENET | | | | |
| MODEL | SCORE | MIN SCORE | PEAKS DIST | MIN PEAKS DIST | COEFF. OF VAR. |
| LPC | $-0.47 \pm 0.00$ | $-0.54 \pm 0.00$ | $10.07 \pm 0.04$ | $9.43 \pm 0.03$ | $0.201 \pm 0.000$ |

$z_i \sim \mathcal{N}\left(\mu_i^{(1,2)}; \sigma_i^{(1,2)^2}\right)$. For each dimension $i$, we build a histogram with the values of all latent representations of the training set. We then fit a bimodal GMM model on this histogram. Assuming that $P$ is the dimensionality of the latent representation and the dataset contains $N$ datapoints, the following quantities are collected: The average log-likelihood score

$$\overline{\ell} = \frac{1}{NP} \sum_{n=0}^{N-1} \sum_{i=0}^{P-1} \log \mathcal{N}\left(z_i^{(n)} \Big| \mu_i^{(1,2)}; \sigma_i^{(1,2)^2}\right); \tag{67}$$

the average standard deviation of the two posterior distributions

$$\overline{\sigma} = \frac{1}{P} \sum_{i=0}^{P} \left(\frac{\sigma_i^{(1)} + \sigma_i^{(2)}}{2}\right); \tag{68}$$

and the mean relative distance of the two peaks reweighted with the standard deviation

$$\overline{\mu} = \frac{1}{P} \sum_{i=0}^{P} \frac{\left\|\mu_i^{(2)} - \mu_i^{(1)}\right\|}{\left(\sigma_i^{(1)} + \sigma_i^{(2)}\right)/2}. \tag{69}$$

We present these values in Table 6. The table shows the average and minimum values for the GMM fitting score and the weighted relative distance between the peaks across all nodes. These three metrics indicate that during training, all latent representations collapse into two distinct points, forming two clearly separated clusters. This observation supports the binarity hypothesis, which states that each latent representation can assume only one of two possible values. For all LPC models the binarity hypothesis holds true for all dimensions, even in cases with the lowest recorded scores. The table also includes the coefficient of variation for the absolute values of the latent representations evaluated for all dimensions and samples. The low values of the coefficient of variation in the LPC architectures indicate that in each node they can assume approximately only one of two possible values. This observation supports our claim that the class means of the latent representations collapse onto the vertices of a hypercube. The same analysis was performed for the LINPEN architecture, which also features a linear layer before classification. However, in this architecture, the binarity hypothesis does not hold.

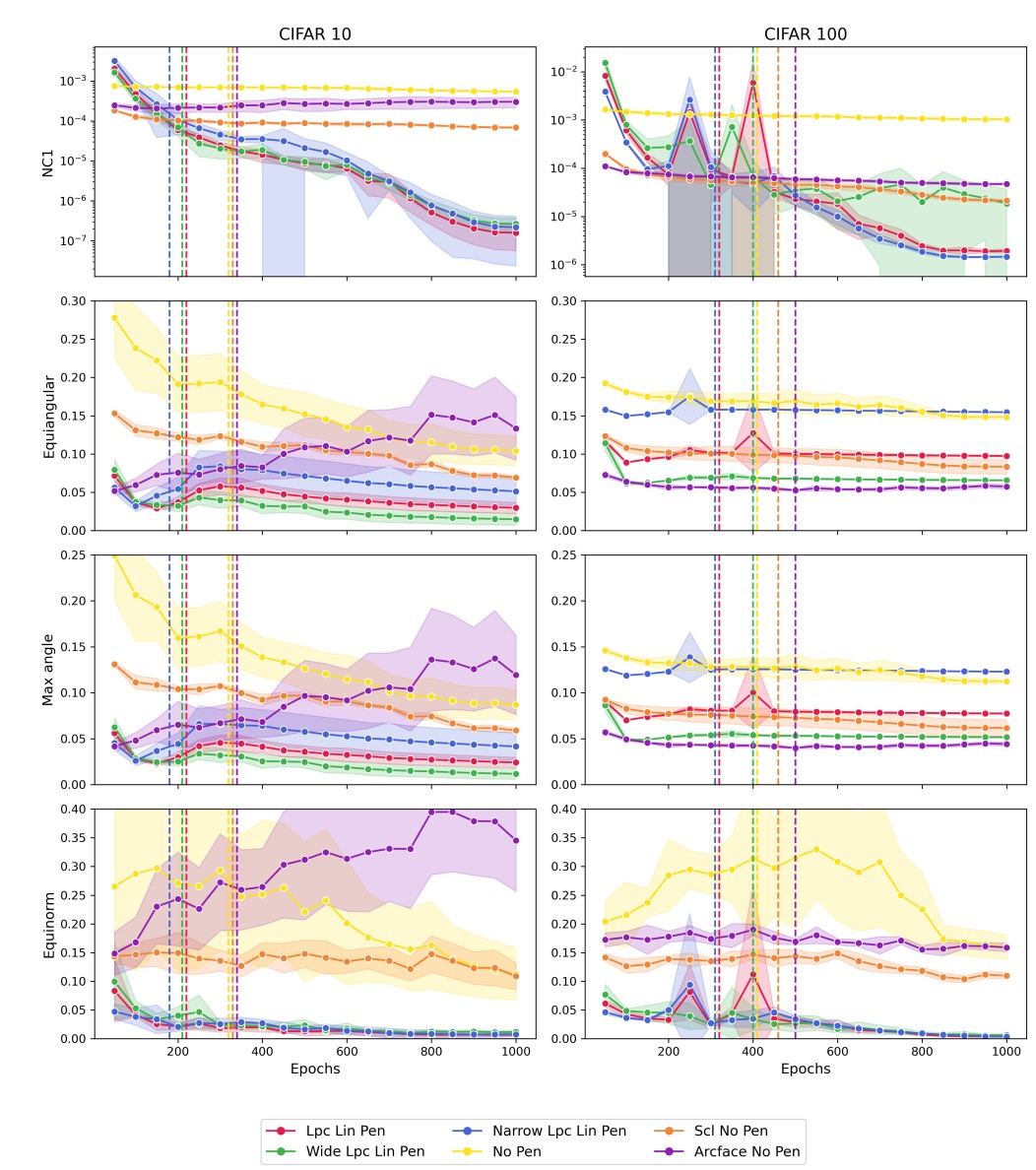

Figure 5: Metrics used to evaluate convergence towards Neural Collapse (NC). In the upper figure, we examine a renormalized version of the NC1 property. This normalization process is conducted based on the number of nodes in the penultimate-layer to ensure a fair comparison across models with varying dimensions of the penultimate-layer. The dashed lines are drawn at the average epoch when training reaches convergence, demonstrating that most of the training was performed in the TPT. Below, we present metrics demonstrating convergence to an ETFS, utilizing the same parameters as those outlined in (11).

# E  NEURAL COLLAPSE

In this appendix, we present all metrics related to NC as defined in (11). The entire NC phenomenon can be summarized into four distinct components: (1) the variability of samples within the same class diminishes as they converge to the class mean (NC1); (2) the class means in the penultimate-layer tend towards an ETFS (NC2); (3) the last layer classifier weights align with the ETFS in their dual

space (NC3); and (4) classification can effectively be reduced to selecting the closest class mean (NC4).

The first property of interest is NC1, which asserts that the variability of samples within the same class decreases in the terminal phase of training. This property is characterized by the equation $\text{Tr}\left(\boldsymbol{\Sigma}_W \boldsymbol{\Sigma}_{\boldsymbol{B}}^{\dagger}/K\right)$, where $\Sigma_W$ is defined as

$$\Sigma_W = \frac{1}{NP}\sum_{i=0}^{N-1}\sum_{p=0}^{P-1}\left(\boldsymbol{z}^{(i,p)}-\boldsymbol{\mu}^{(p)}\right)\left(\boldsymbol{z}^{(i,p)}-\boldsymbol{\mu}^{(p)}\right)^{\top} \tag{70}$$

where $\boldsymbol{z}^{(i,p)}$ is the $i$-th latent representation with label $p$ and, $\boldsymbol{\mu}^{(p)}$ is the mean of all representation with label $p$; and $\Sigma_B$ is defined as:

$$\Sigma_B = \frac{1}{P}\sum_{p=0}^{P-1}\left(\boldsymbol{\mu}^{(p)}-\boldsymbol{\mu}_G\right)\left(\boldsymbol{\mu}^{(p)}-\boldsymbol{\mu}_G\right)^{\top} \tag{71}$$

where $\boldsymbol{\mu}_G$ represents the global mean of all class means. The trace operation sums over all diagonal elements, the dimensionality of which is equal to that of the penultimate-layer, $P$. Given the use of different architectures with varying numbers of nodes in the penultimate-layers in our study, we examine a renormalized version of this quantity, $\text{Tr}\left(\boldsymbol{\Sigma}_W \boldsymbol{\Sigma}_{\boldsymbol{B}}^{\dagger}/K/P\right)$.

The second property, NC2, characterizes the convergence of class means to a Simplex Equiangular Tight Frame (ETF). We evaluate this convergence using three key metrics shown in the lower panels of Fig. 5. The first metric is the equinorm property, which measures how uniform the norms of the centered class means become:

$$\text{Equinorm} = \frac{\text{Std}_p(\|\boldsymbol{\mu}^{(p)}-\boldsymbol{\mu}_G\|_2)}{\text{Avg}_p(\|\boldsymbol{\mu}^{(p)}-\boldsymbol{\mu}_G\|_2)} \tag{72}$$

where $\text{Std}_p(\cdot)$ is the standard deviation across classes, and $\text{Avg}_p(\cdot)$ is the average across classes. As training progresses, this value approaches zero, indicating that all class means have approximately equal norms.

The second metric is the equiangularity property, which measures how uniform the angles between different pairs of class means become:

$$\text{Equiangularity} = \text{Std}_{p,p'\neq p}(\cos_{\boldsymbol{\mu}}(p,p')) \tag{73}$$

where $\cos_{\boldsymbol{\mu}}(p,p') = \left\langle \boldsymbol{\mu}^{(p)}-\boldsymbol{\mu}_G, \boldsymbol{\mu}^{(p')}-\boldsymbol{\mu}_G\right\rangle/(\|\boldsymbol{\mu}^{(p)}-\boldsymbol{\mu}_G\|_2\|\boldsymbol{\mu}^{(p')}-\boldsymbol{\mu}_G\|_2)$. As training progresses, this value approaches zero, indicating that all pairs of class means form equal angles.

The third metric is the maximal-angle equiangularity, which measures how close the angles between class means are to their theoretical optimal value in an ETFS:

$$\text{Maximal-Angle} = \text{Avg}_{p,p'\neq p}|\cos_{\boldsymbol{\mu}}(p,p')+1/(P-1)| \tag{74}$$

In an ideal ETFS, all cosines should equal $-1/(P-1)$, which represents the maximum separation possible for globally centered, equiangular vectors. As training progresses, this value approaches zero, indicating optimal angular separation.

In Fig. 5, the top image presents the normalized NC1 value, showing that it is orders of magnitude lower in the LPC architectures compared to the baseline architecture. We also note that the other regularization techniques SCL and ArcFace provide better convergence to NC with respect to the baseline, but improvements remain lower with respect to LPC models.

The other three images below demonstrate the convergence of class means toward an ETFS (NC2 property). These images show that all values reach a plateau in the terminal phase, indicating convergence to their optimal values. It is evident that the LPC-NARROW architecture, which uses a smaller-dimensional embedding in the penultimate-layer, tends to exhibit higher values for the angular measures (maximum angle and equiangularity) compared to the baseline. This is because, geometrically, it is more challenging for the network to construct an ETFS using the vertices of a hypercube in a low-dimensional space.

By observing the metrics in Fig. 5, we conclude that while regularization techniques accelerate convergence to NC, the best convergence is achieved with LPC. We also note that the dashed lines represent the average epoch at which the network reached convergence, showing that most of the training occurred after convergence, in the TPT. All metrics have reached plateaus, demonstrating that the phenomenon of NC is fully realized. Thus, the additional benefits of LPC documented in this paper are in addition to those typically associated with NC.

## F   LLMs Usage

In the preparation of this manuscript, Large Language Models (LLMs) were utilized as an assistive tool to enhance the quality and presentation of our work. The primary applications of the LLMs were for text polishing and improving overall clarity and readability. Additionally, the LLM played a role in more technical aspects of manuscript preparation, specifically in the completion and formatting of tables. The authors reviewed, edited, and take full responsibility for all content, including any contributions from LLMs, to ensure the scientific integrity and accuracy of this paper.

