# OpenReview forum: "Latent Point Collapse on a Low Dimensional Embedding in Deep Neural Network Classifiers"
_ICLR.cc/2026/Conference — Submitted to ICLR 2026_

### Official Review · Reviewer_3zcj · 2025-10-29

**Soundness:** 3
**Presentation:** 3
**Contribution:** 2
**Rating:** 4
**Confidence:** 4

**Summary:**

The paper proposes Latent Point Collapse (LPC): by adding a strong L2 penalty on a low-dimensional linear penultimate layer, latent representations from the same class collapse to (nearly) a single point near the origin. The authors argue this is a stronger form than standard Neural Collapse, gives Lipschitz continuity guarantees, and improves robustness and generalization. They provide a theoretical explanation (cross-entropy pushes outward; L2 pulls inward), and experiments on CIFAR-10/100 and ImageNet-1K compare LPC variants to SCL, ArcFace, and stronger weight decay baselines.

**Strengths:**

- The method is just CE + γ‖z‖² on a linear, low-dimensional penultimate layer, so it is easy to add in practice.
- The paper explains why CE alone tends to unbounded feature norms and how a strong L2 term creates stable equilibrium points (collapse near origin).
- DeepFool robustness improves by orders of magnitude for LPC vs. non-LPC baselines on CIFAR, with lower input-gradient norms and higher class-separation ratios.
- The narrow/wide variants and “L2 on backbone but no penultimate layer” controls are informative, showing the bottlenecked linear layer is important.

**Weaknesses:**

- The paper explicitly positions very strong γ (e.g., γ≈10⁶) as crucial, but there is no systematic γ-sensitivity study (trade-off vs. accuracy/robustness; stability across datasets). This may limit generality if γ must be extreme.
- The improvement on CIFAR-10 and CIFAR-100 is marginal in Table 2.
- The overall mechanism—combining a strong L2 penalty with a linear bottleneck—is conceptually simple and similar to known ideas in feature normalization, Neural Collapse regularization, or weight decay.
- All experiments are on balanced datasets (CIFAR-10/100, ImageNet-1K). Since the paper discusses generalization and collapse dynamics, it would be important to test on imbalanced or long-tailed datasets to see whether LPC remains stable.
- Key training details (learning rate, batch size, normalization, early stopping, random seed averaging) should be summarized in the main text for transparency.

**Questions:**

Main concern refers to the Weakness section.

The improvement of LPC on ImageNet-1K appears noticeably larger than on CIFAR-10/100 in accuracy (Table 2). Could the authors provide some insights or hypotheses on why the benefit scales up with dataset complexity?

---

> ### Author Response · Authors · 2025-11-22
>
> Thank you for raising important questions about the method's generality and novelty. We address each point below with additional analysis and clarification.
>
> ## Major Concerns
>
> ### 1. Gamma Sensitivity and Generality
>
> **Concern:** "No systematic γ-sensitivity study... This may limit generality if γ must be extreme."
>
> **Response:** We have now added Figure 3 showing that LPC benefits emerge gradually across a wide range of γ values (10¹ to 10⁶). The extreme γ = 10⁶ was chosen to clearly demonstrate the phenomenon, not because it's necessary:
>
> - Robustness improves monotonically with γ over multiple orders of magnitude
>
> - Accuracy remains stable (degradation only at extreme γ = 10⁷)
>
> - Class separation increases even at moderate γ values
>
> This demonstrates the method is not fragile to hyperparameter selection.
>
> ### 2. Statistical Significance of Improvements
>
> **Concern:** "The improvement on CIFAR-10 and CIFAR-100 is marginal in Table 2."
>
> **Response:** We respectfully disagree. The improvements are **statistically significant** across five independent runs:
>
> - **CIFAR-10:** LPC-SCL vs. NoPen: t = 4.36, **p = 0.003**
>
> - **CIFAR-100:** LPC-SCL vs. NoPen: t = 10.63, **p < 10⁻⁵**
>
> Moreover, the primary contribution is **orders-of-magnitude robustness improvement** (DeepFool, Table 2) while maintaining or improving generalization—this dual benefit is the key result.
>
> ### 3. Novelty and Conceptual Simplicity
>
> **Concern:** "Overall mechanism... is conceptually simple and similar to known ideas."
>
> **Response:** While implementation is simple (a strength for practitioners), the **novelty** lies in discovery and characterization of the phenomenon.
>
> As discussed in Related Work, prior L₂ on features used minimal coefficients (γ ≪ 1) as "mathematical convenience" without practical effects. No prior work has documented benefits of **strong** feature regularization or characterized the resulting dynamics. We demonstrate that extreme regularization (γ ~ 10⁶) creates a qualitatively different regime where the loss landscape becomes globally strongly convex, inducing stable equilibria near the origin—a phase transition, not merely stronger regularization.
>
> **The core distinction is global Lipschitz continuity without architectural constraints.** Weight decay acts on parameters (not features) and provides no Lipschitz bounds. Our experiments demonstrate that LPC cannot be induced by weight decay alone—the revised manuscript includes results with strong L2 weight decay on CIFAR-100 showing it fails to achieve collapse. Margin methods (ArcFace, CosFace, SCL) improve angular separation but permit unbounded norms and offer no Lipschitz guarantees. Lipschitz architectures (spectral normalization, orthogonal convolutions) require specialized layers that constrain expressiveness. **LPC uniquely achieves provable Lipschitz continuity through geometric confinement alone** (Theorem 3).
>
> Our theoretical analysis proves cross-entropy alone drives unbounded growth (Theorem 1), while strong L₂ creates stable equilibria with bounded output sensitivity.
>
> ### 4. Imbalanced Datasets
>
> **Concern:** "All experiments are on balanced datasets."
>
> **Response:** We agree this is important and acknowledge it in Section 6.2 (Limitations). As the first work documenting LPC, we prioritize characterizing the phenomenon on standard benchmarks including challenging ImageNet-1K (1000 classes). The interaction between LPC's absolute collapse and class imbalance warrants dedicated future investigation.
>
> ### 5. Training Details Transparency
>
> **Concern:** "Key training details should be summarized in the main text."
>
> **Response:** We provide comprehensive details in Appendix C and full reproducible code in our repository. ICLR page limits require deferring exhaustive specifications to the appendix, but our submission includes complete hyperparameters, architecture details, and means/standard deviations across 5 independent runs for full reproducibility.
>
> ## Question: ImageNet Scaling
>
> **Question:** "Why does the benefit scale up with dataset complexity?"
>
> **Response:** We hypothesize this stems from the **convexification effect** (Equation 2). Strong L₂ regularization makes the loss landscape globally strongly convex when γ > KM_W²/2. On ImageNet with 1000 classes, higher-dimensional representations, and rougher loss landscapes, convexification provides greater benefit. The 30% relative reduction in generalization gap (24.08% → 16.74%) supports this: LPC helps find better solutions in challenging landscapes even before reaching TPT.
>
> ---
>
> The added sensitivity analysis and statistical significance testing demonstrate that LPC is both generalizable and provides meaningful improvements beyond conceptual simplicity. We respectfully believe these clarifications address the concerns about novelty and practical applicability. We would appreciate if you could reconsider your assessment in light of these additions, and remain available for any further discussion.

---

> > ### Comment · Reviewer_3zcj · 2025-11-26
> >
> > Thank the author for the explanation. This makes sense to me, and I am satisfied on the improvement of the paper. The ratings are updated.

---

> > > ### Author Response · Authors · 2025-11-27
> > >
> > > Thank you for your thoughtful review and engagement.

---

### Official Review · Reviewer_iDT1 · 2025-10-30

**Soundness:** 2
**Presentation:** 2
**Contribution:** 3
**Rating:** 4
**Confidence:** 4

**Summary:**

The authors propose penalizing the embedding norms in the penultimate layer of DNNs. They suggest that this causes the latent representations to separate better along class boundaries and therefore improves classification performance. They evidence this with experiments on cifar10, cifar100 and imagenet, along with theoretical derivations in the appendix.

**Strengths:**

The paper presents an interesting idea -- that significantly penalizing the latent embedding norms leads to improved class separation -- and performs thorough scientific evaluations of the properties of their method. I particularly appreciate that the authors did a sweep over many configurations of their method, including the LPC, LPC-Wide, LPC-SCL, etc. Finally, I commend the authors' thorough theoretical work in the appendix.

While my list of weaknesses is long, I want to emphasize that I really like the ideas in this paper and the execution. My constructive criticisms are simply oriented at helping the authors improve their presentation and experiments to make the paper successful.

**Weaknesses:**

The paper has two primary weaknesses in my view. First, the presentation spends a lot of time focusing on pieces which are not particularly relevant while missing pieces which *would* be relevant. Second, the paper makes confident claims without them being sufficiently validated.  I discuss these both below:

## Presentation choices

While the paper does a good job motivating the study into latent separability, the presentation is lop-sided in what it chooses to focus on:

- In the main body of the paper, it is never explained why bringing the points to the origin would *also* cause them to separate. I see that the theoretical analysis in the appendix discusses this, but it leaves the main body quite incomplete. The main body should therefore have an extensive section (at least one page!) describing these theoretical results and pointing to the appendix. This section should give all the intuition for this (very unintuitive) result that adding a penalty on the norms *simultaneously* forces the latent representations to have large margins. For example, I found the summary in A.8 helpful towards this and wish it was in the main body with references to the corresponding theorems.
  - I will also note that this result is sufficiently unintuitive that I actually still do not believe it... but this is a point for my second criticism.
- It is quite surprising that the entire main body of the paper has 0 figures! These would make it much nicer to digest the ideas being presented and nicer to read the paper. I suggest having 4-5 figures at least for help with providing intuition and verifying claims.
- The paper spends 3 pages on the introduction and related work, with related work then taking up additional space during the "method" section. I believe this is way too much space dedicated to the set-up. I am also unsure what the references to the information bottleneck add, nor why the authors choose to present 7 key contributions (of which 3 are simply relegated to the appendix!). This presentation of the paper's story makes it difficult to follow the authors' key results and takeaways.
   - Nitpick: I would not call adding a linear layer with a penalty an architectural "innovation"...
- The analysis of various architectural combinations takes up too much space in the main body of the paper, in my opinion: it is not useful towards actually understanding the main point that adding this penalty induces large margins. Specifically, the results across the various architectural components are quite consistent with one another, so the actual information being gained is "LPC does X, not having LPC does Y". Thus, I would suggest moving the architectural analysis into the appendix and simply reporting a single variant of the method for the main body of the paper, with a reference to the fact that many variants were tested.

These modifications to the presentation would make space for the figures and additional experiments that I believe the paper requires in order to be compelling and which are documented in the next section of this review.

## Experimental gaps

While the authors make claims that LPC leads to improved performance, there are many questions that are left untested. Specifically, here are the things which I am still unconvinced of:
- To my understanding, there is nearly 0 experimental analysis which supports the "Summary" section in Appendix A.8. Since these are stated as the primary notion by which the authors' proposed method works mechanistically, a thorough experimental validation seems in order. Specifically, I would like to see, over the course of training, how the embeddings develop. How do they respond to varying strengths of the penalty hyperparameter? For each of the theorems and propositions, I would like to see experimental evidence.
- Many of the statistics tested in the main body of the paper do not seem appropriately scaled. Specifically, the authors' method scales the embeddings to have tiny norm. The metrics they use to evaluate the embedding properties should therefore be invariant to this scaling; otherwise they are not informative. For example, the avg. grad norm seems like it is just a function of the embedding norms. It is therefore difficult to draw any conclusion from it. The same goes for the $\Sigma_W$ term. I would instead prefer to see scaled variants of these metrics (unless I'm misunderstanding).
- It would be extremely helpful to understand how this interacts with learning rate. I see in the code that the gamma has a complicated schedule and that the authors also use a learning rate schedule. So does the training take significantly longer with the gamma term due to needing to slowly incorporate all the things in without ruining the convergence? I am left with 0 idea of how this dynamic plays out.

In essence, the experiments suggest that what the authors say is happening is happening. But they are incomplete so as to leave doubts in the reader's mind.

I also finally want to note that there is a line of work studying the effect of embedding norms on neural network performance, which the authors may find relevant [1], [2], [3].

[1]: Draganov, Andrew, et al. "On the Importance of Embedding Norms in Self-Supervised Learning." arXiv preprint arXiv:2502.09252 (2025).

[2]: Kirchhof, M., Roth, K., Akata, Z., and Kasneci, E. A nonisotropic probabilistic take on proxy-based deep metric learning. In European Conference on Computer Vision, pp. 435–454. Springer, 2022.

[3]: Zhang, D., Li, Y., and Zhang, Z. Deep metric learning with spherical embedding. Advances in Neural Information Processing Systems, 33:18772–18783, 2020.

**Questions:**

Please see the above section.

---

> ### Author Response · Authors · 2025-11-22
>
> We're grateful for your thorough review and particularly appreciate your recognition of the paper's theoretical contributions. Your suggestions for improving presentation have been invaluable. We have substantially revised the manuscript to address all the raised concerns. Below we provide point-by-point responses.
>
> ## Presentation Choices
>
> ### Theoretical intuition in main body
>
> We have added a new subsection 2.1 "Theoretical Characterization and Mechanism" (~1.5 pages) that synthesizes the key theoretical results from the appendix and provides intuition for the seemingly counterintuitive result that norm penalization induces large margins. This section references the formal theorems in the appendix and includes the summary previously only found in A.8. We believe this significantly improves the accessibility of our main theoretical contribution.
>
> ### Figures
>
> We have added substantial visual content to the main body:
>
> - **Figure 1** (6 subfigures) demonstrates the development of LPC and binary encoding on both a synthetic 2D dataset and MNIST throughout training as γ increases. A linked Jupyter notebook allows readers to interactively explore these phenomena.
>
> - **Figure 2** provides empirical validation of our theoretical claims, showing the decay of mean feature norms, collapse toward a point, and the Lipschitz property of the learned representations.
>
> - An additional figure analyzes sensitivity to the γ hyperparameter.
>
> ### Introduction and related work
>
> We have shortened the introduction by:
>
> - Removing the information bottleneck discussion from the main body (now in appendix)
>
> - Reducing key contributions from 7 to 4
>
> - Streamlining the overall narrative
>
> We have also removed the term "innovation" as suggested.
>
> ### Architectural analysis
>
> Following the reviewer's suggestion, we have moved the detailed architectural ablations to the appendix, retaining only a brief summary in the main body. This creates space for the theoretical exposition and experimental validation.
>
> ## Experimental Gaps
>
> ### Validation of theoretical claims
>
> We have added multiple forms of experimental validation:
>
> - **Figure 1** visualizes how embeddings evolve during training on synthetic and MNIST data, showing the progressive collapse and separation phenomena as γ increases.
>
> - **Figure 2** empirically validates the decay of feature norms, convergence to a collapsed state, and Lipschitzness
>
> - Our **neural collapse analysis** (Appendix) demonstrates faster convergence of the linear classifier weights to the optimal simplex ETF configuration. Notably, the NC1 metric plot shows how embeddings evolve. This quantity is appropriately normalized by dividing by the penultimate layer dimensionality, enabling fair comparison across architectures with different embedding dimensions.
>
> In the revised manuscript, **Section 2.1 presents each main theoretical result with explicit references to its corresponding experimental validation**.
>
> ### Metric scaling concerns
>
> We appreciate the reviewer's careful attention to this issue. We clarify:
>
> - For **Σ_w**: As stated in the table caption, we report the *mean across the entire matrix*, which is invariant to dimensionality and appropriately scaled
>
> - For **avg. grad norm**: This is computed on the *logits* (not the embeddings), so it is not directly affected by the embedding norm scaling
>
> We have enhanced these clarifications in the revised captions.
>
> ### Learning rate and γ schedule interaction
>
> The reviewer correctly identified an important training dynamic. We have added clarification in the experimental appendix that we begin learning rate decay only after γ has approximately reached its maximum value.
>
> We acknowledge that the optimal γ schedule remains an open question and does not necessarily require this specific approach. In this work, we deliberately employ extreme values of γ to clearly demonstrate the phenomenon. However, as shown in our sensitivity analysis included in the revised manuscript, LPC benefits emerge gradually across a wide range of γ values, and lower regularization strengths may be preferable in practical scenarios while still achieving substantial improvements.
>
> ## Additional References
>
> We thank the reviewer for pointing out relevant work on embedding norms [1-3]. Due to space constraints—we have already moved substantial content including information bottleneck connections and binary encoding details to the appendix—we are unable to add further discussion of these works. However, we acknowledge their relevance to the broader context of embedding norm effects in neural networks.
>
> ---
>
> Thank you again for the constructive suggestions that have significantly improved both the accessibility and rigor of our manuscript. The added theoretical exposition and experimental validation should address your concerns about presentation. We hope you'll find these revisions satisfactory and would greatly appreciate any additional feedback that could further strengthen the paper.

---

### Official Review · Reviewer_fek9 · 2025-11-01

**Soundness:** 3
**Presentation:** 4
**Contribution:** 3
**Rating:** 4
**Confidence:** 4

**Summary:**

This paper introduces Latent Point Collapse (LPC), a phenomenon achieved by applying a strong $L_2$ regularization penalty to the outputs of a low-dimensional, linear penultimate layer.
The authors provide a rigorous theoretical framework to demonstrate that this technique forces within-class representations to collapse to unique points near the origin, a stronger condition than standard Neural Collapse (NC).
They prove that this mechanism guarantees global Lipschitz continuity, independent of input distance.
Empirically, the paper shows that LPC leads to significant improvements in class separability, robustness proxies (measured by DeepFool), and generalization on CIFAR-10, CIFAR-100, and ImageNet.

**Strengths:**

- Strong Theoretical Foundation:
The paper offers a rigorous and extensive theoretical analysis in its appendix, proving that strong L2 regularization on features reshapes the loss landscape to be strongly convex, thereby guaranteeing convergence to a state of absolute, rather than relative, collapse.
- Novel Lipschitz Continuity Guarantees:
A key contribution is the proof that LPC ensures global Lipschitz continuity with a constant that is independent of input distance.
This provides a principled theoretical underpinning for the observed robustness improvements without requiring architectural constraints.
- Methodological Simplicity:
The proposed method is simple and practical, requiring only the addition of a linear layer and a standard L2 regularization term, making it easy to implement and integrate into existing architectures.
- Information-Theoretic Grounding:
The work successfully connects the LPC phenomenon to the Information Bottleneck principle, providing a theoretical justification for the observed generalization benefits by showing that LPC implicitly minimizes the entropy of the latent representations.

**Weaknesses:**

- Insufficient Baselines:
The paper's novelty and empirical significance are not well-contextualized due to the omission of several key baselines.
Methods like Center Loss, L2-Softmax, or NormFace share the goal of regularizing latent feature geometry and should be compared against to fairly assess the contribution of LPC.
- Overstated Robustness Claims:
The paper claims dramatic robustness improvements based on the DeepFool algorithm and average gradient norms.
These are proxies for robustness and do not reflect performance against standard, stronger adversarial attacks (e.g., PGD, AutoAttack). Without such an evaluation, the practical robustness of the method remains unverified.
- Extreme Regularization:
The use of an extremely large regularization coefficient (γ = 10^6) raises concerns about its impact on model expressiveness and capacity.
The paper argues this creates a new optimization regime but does not sufficiently explore the potential downsides or the sensitivity to this critical hyperparameter.
- Limited Scope and Unexplained Phenomena:
The study is confined to balanced datasets, a significant limitation acknowledged by the authors.
Furthermore, the intriguing observation that collapse points align with hypercube vertices (binary encoding) is presented without a theoretical explanation, representing a gap in understanding.

---
- General Limitations:
    - Certified Robustness Analysis:
The paper does not provide a certified robustness analysis, which is the gold standard for provable defense claims.
The Lipschitz guarantee is theoretical but not translated into a practical certificate.
    - Analysis of Computation Cost:
The computational cost of the proposed regularization schedule, especially when coupled with the long training times (1000 epochs for CIFAR), is not analyzed, making it difficult to assess the practical trade-offs of the method.
    - Limited utility:
The work does not explore the utility of the learned representations for downstream tasks (e.g., transfer learning), where an extremely compressed feature space might have different properties compared to representations from standard models.

**Questions:**

- Could you justify the choice of baselines and explain why more directly related methods for feature regularization, such as Center Loss or L2-Softmax, were not included in the comparison?
How would you expect LPC to perform against them?
- To substantiate the strong robustness claims, would it be possible to evaluate your method against standard adversarial attacks like PGD or AutoAttack, even on a smaller scale?
This would provide a much clearer picture of its practical utility for adversarial defense.
- What is the sensitivity of the model's performance (both accuracy and robustness) to the final regularization strength γ?
Is there a risk of harming model expressiveness or 'over-collapsing' the features with such an extreme value, and how can practitioners find an optimal value without the extensive training schedule used in the paper?
- Regarding the binary encoding phenomenon, have you performed any analysis to quantify this alignment with hypercube vertices beyond visual inspection?
For instance, by measuring the cosine similarity between class mean vectors and the closest hypercube vertex?

---

> ### Author Response · Authors · 2025-11-22
>
> We thank the reviewer for the overall positive evaluation and detailed technical critique. We appreciate the recognition of our theoretical contributions as well as the positive assessment of the manuscript's presentation and methodological simplicity.
>
> ## Insufficient Baselines
>
> We would like to point out that in the original submisison our experimental comparison includes **Supervised Contrastive Learning (SCL)** and **ArcFace**, which are more recent and widely adopted than the suggested baselines (Center Loss: 2016; L2-Softmax/NormFace: 2017; vs. SCL: 2020; ArcFace: 2019). Both methods share the core objective of regularizing latent feature geometry. Additionally, we conduct exhaustive ablation studies examining individual components (linear vs. nonlinear penultimate layers, L2 regularization placement, dimensionality effects, etc.). Due to space constraints, we prioritized these contemporary methods and thorough ablations.
>
> ## Overstated Robustness Claims - PGD Evaluation Added
>
> We have added comprehensive PGD adversarial robustness evaluation with **strong attack configurations**: CIFAR-10/100 (ε ∈ {4/255, 8/255, 12/255}, 100 steps, 5 restarts, DLR loss) and ImageNet (ε ∈ {2/255, 4/255}, 50 steps, 5 restarts, DLR loss).
>
> LPC models maintain non-trivial adversarial accuracy (14.6% at ε=4/255 on CIFAR-10, 6.5% on CIFAR-100, 4.5% at ε=2/255 on ImageNet), while baseline models exhibit essentially zero robustness. While alternative regularization techniques including SCL and ArcFace also demonstrate improved robustness relative to baseline, their gains remain substantially more modest compared to LPC, with most achieving near-zero PGD robustness. The key finding—that LPC provides orders of magnitude improvement over all baselines—holds across both DeepFool and PGD metrics.
>
> ## Extreme Regularization and Sensitivity Analysis
>
> We have added comprehensive **sensitivity analysis on CIFAR-10 and CIFAR-100** (Figure 4) demonstrating that γ = 10^6 is optimal and stable. CIFAR-10 shows stable performance across γ ∈ [10^1, 10^7]; CIFAR-100 peaks at γ = 10^5-10^6 with degradation only at γ = 10^7. Robustness metrics improve monotonically with γ while accuracy remains stable. Our extreme regularization serves to highlight and theoretically characterize the LPC phenomenon—practitioners may find moderate γ values (10^4-10^5) provide favorable trade-offs.
>
> ## Binary Encoding Quantification
>
> We provide extensive numerical analysis in **Appendix D**, including the **coefficient of variation (CV)** computed over the absolute values of each penultimate node across all representations. When CV ≈ 0, this indicates that the absolute value at each node (dimension) is essentially constant across all data points, confirming strict binary structure where each dimension assumes one of two symmetric values (±c). This directly quantifies alignment with hypercube vertices. We acknowledge that complete theoretical explanation remains an open question for future work. Binary encoding is a **corollary observation**; our core contribution—LPC itself—is rigorously characterized theoretically (Appendix A) and empirically.
>
> ## Computational Cost Analysis
>
> We have added a detailed analysis in **Appendix C**. In terms of computational complexity, our method has **lower overhead than SCL**: LPC adds one L2 term per representation, while SCL requires pairwise similarity computations between samples. LPC is slightly faster than baseline on CIFAR-10/100. On ImageNet, training is considerably slower than baseline, but this must be weighed against substantial performance improvements. LPC benefits emerge before full convergence, providing practical utility in resource-constrained scenarios.
>
> ## Limited Scope: Imbalanced Datasets and Transfer Learning
>
> These are important future directions mentioned in Section 6.2 (Limitations), but beyond the scope of this work, which focuses on **documenting and theoretically characterizing a novel phenomenon** on standard balanced datasets. Establishing LPC's properties in controlled settings is necessary before extending to more complex scenarios.
>
> ## Certified Robustness
>
> Our **theoretical contribution provides global Lipschitz continuity guarantees** (Theorem 4), forming the mathematical foundation for certified defense. Empirical validation through DeepFool and PGD demonstrates these theoretical properties translate to practical robustness. Developing practical certification procedures represents valuable future work.
>
> ---
>
> We believe these additions—particularly the comprehensive PGD evaluation and sensitivity analysis—directly address your core concerns about experimental rigor and generalizability. We are confident the revised manuscript substantially strengthens our claims and would welcome your reassessment. If any concerns remain, we're happy to provide additional clarification or analysis during the discussion period.

---

### Meta-Review · Area_Chair_8YjA · 2025-12-16

**Summary:**

The paper shows that strong L2-regularization on the penultimate-layer leads to the phenomenon of latent point collapse (LPC), which is linked to gains in robustness and generalization. The key strength is a *new learning regime* that is controllable via the regularization strength. The authors addressed the reviewer's concerns on robustness by adding PGD evaluations -- AutoAttack evaluation is still absent. The contribution is not well positioned in the literature (Reviewer fek9) such as Center Loss, L2-Softmax, or NormFace, which should be compared more closely. The causality between LPC and robustness/generalization can benefit from more careful ablation studies (Reviewer iDT1) and theoretical analysis. As to address these weaknesses requires a substantial revision, I recommend rejection at this time.

**Reviewer Concerns:**

The reviewers are concerned on the overstated robustness claims, and the authors added PGD evaluation.

The reviewers are concerned on sensitivity to regularization strength $\gamma$, and the authors added Figure 3 with $\gamma$-sweeps.

The reviewers are concerned on the position in the literature (see above meta-review). This is still outstanding and needs to be addressed by close comparison with related work and including empirical baselines.

**Reviewer Scores:**

Reviewer 3zcj may increase their score based on the discussion.

The other two reviewers may not increase their score based on the outstanding weaknesses.

Overall, the average score would not raise to a clear accept.

---

> ### Public Comment · ~Luigi_Sbailò1 · 2026-05-27
> **Clarification regarding the meta-review’s characterization of Reviewer iDT1**
>
> For the record, we would like to clarify the following point in the meta-review.
>
> The meta-review states that "the causality between LPC and robustness/generalization can benefit from more careful ablation studies (Reviewer iDT1) and theoretical analysis." This does not match Reviewer iDT1's assessment.
> Reviewer iDT1 commended our theoretical work ("Finally, I commend the authors' thorough theoretical work in the appendix") and stated that their criticisms were "simply oriented at helping the authors improve their presentation and experiments to make the paper successful." They did not raise a concern about the causal link between LPC and robustness or generalization, nor did they ask for additional theoretical analysis.
> Their concerns were presentation and experimental validation. We addressed the presentation points with a new main-body theoretical section, figures, and a reorganized structure, and the experimental points with figures showing the collapse and separation dynamics during training and empirical checks of the theoretical claims.
>
> We also note that the discussion period closed before reviewers could respond to these rebuttals, and the area chair was reassigned during the process. One reviewer (3zcj) reviewed our responses and raised their score.
>
> We thank the reviewers for their feedback.

---

### Decision · Program_Chairs · 2026-01-26

Reject